# Eye tracking and eye expression decoding based on transparent, flexible and ultra-persistent electrostatic interface

Yuxiang Shi [1,2], Peng Yang[1,2], Rui Lei[1], Zhaoqi Liu[1,2], Xuanyi Dong[1,2], Xinglin Tao[1,2], Xiangcheng Chu[3], Zhong Lin Wang[1,4] & Xiangyu Chen [1,2] ✉

Eye tracking provides valuable insight for analyzing visual attention and underlying thinking progress through the observation of eye movements. Here, a transparent, flexible and ultra-persistent electrostatic sensing interface is proposed for realizing active eye tracking (AET) system based on the electrostatic induction effect. Through a triple-layer structure combined with a dielectric bilayer and a rough-surface Ag nanowire (Ag NW) electrode layer, the inherent capacitance and interfacial trapping density of the electrostatic interface has been strongly enhanced, contributing to an unprecedented charge storage capability. The electrostatic charge density of the interface reached 1671.10 $\mu C \cdot m^{-2}$ with a charge-keeping rate of 96.91% after 1000 non-contact operation cycles, which can finally realize oculogyric detection with an angular resolution of 5°. Thus, the AET system enables real-time decoding eye movements for customer preference recording and eye-controlled human-computer interaction, supporting its limitless potentiality in commercial purpose, virtual reality, human computer interactions and medical monitoring.

Eye tracking, a revolutionary technique that can provide information about human visual behaviors[1,2], attentional processes[3,4] and even decision processes[5,6] by decoding eyeball movements, points of gaze and blinks, has tremendous applications in medical[7,8], commercial[9,10] and engineering[11,12] fields. For instance, eye trackers have been applied in the rehabilitation of cognitive function[13], assistant for people suffering from amyotrophic lateral sclerosis (ALS)[14], such as Stephen William Hawking, defining consumer's product preference[10,15] and featuring a visual reality (VR) system as an approach for human-computer interactions. Therefore, the development of eye tracking system is particularly important for the prosperity of eye-based health monitoring, human–machine engineering improvements and commercial analysis. To date, many different methods have been proposed to realize eye tracking, including scleral search coil[16,17], magnetic resonance (MR)[18,19], video oculography[13,20] and oculography[21,22]. However, there are still unsolved problems for eye tracking techniques. The scleral search coil requires an invasive coil in the eye[23] which can cause eye infection and cannot avoid slipping as the eye rotates. MR trackers rely independently on cumbersome equipment[24], leading to insufficient portability. As for the video oculography, high resolution is a notable advantage based on infrared light mirrored from sclera, but it is fairly limited for portability because of privacy concerns and awkward location of the camera[25,26]. The electrooculography (EOG) sensor is also a promising approach based on electrical signals. It theoretically owns high resolution based on the eye dipole with a positive cornea and negative retina, and has been applied in human-machine interfaces and diagnosis[27,28]. However, limited breathability and infectious risks of contact-mode electrodes (usually containing Ag/AgCl) may induce

[1]CAS Center for Excellence in Nanoscience, Beijing Key Laboratory of Micro-nano Energy and Sensor, Beijing Institute of Nanoenergy and Nanosystems, Chinese Academy of Sciences, Beijing 100083, China. [2]School of Nanoscience and Engineering, University of Chinese Academy of Sciences, 100049 Beijing, China. [3]State Key Laboratory of New Ceramics and Fine Processing, Tsinghua University, Beijing 100084, China. [4]Georgia Institute of Technology, Atlanta, GA 30332-0245, USA. ✉e-mail: chenxiangyu@binn.cas.cn

discomfort feeling[29,30]. Hence, it is still critical to explore new method based on electric signal for eye tracking.

Electrostatic induction is a well-known phenomenon that roots from hundreds of years ago and has attracted great interest in recent decades mainly due to the rapid development of triboelectric nanogenerator (TENG). TENG-based wearable sensors have been utilized for chemical or biological detection due to their low cost, high sensitivity and multimode operation[31–34]. The static charges on the dielectric film of TENG can generate electrostatic field in the surrounding regions, and moving objects, such as charged body and conductor, are able to induce varied intensities of the field[35], thus giving rise to detectable electrical signals. TENG bears abilities to transfer low-frequency irregular mechanical energy into electricity, which can reduce the complexity of motion detection and contribute to an active non-contact sensor system[36]. Hence, TENG has been applied for gesture recognition and motion mapping in various sensing areas[37–41], and also has great potential to be used in eye tracking area.

In this article, we report an active eye-tracking (AET) system for real-time eye movements monitoring and eye-controlled human-computer interaction. The working principle of this AET system is similar to a non-contact TENG and it depends on a transparent and flexible electrostatic interface attached on glasses, which is composed of an electrostatic-charged composite dielectric bilayer and pre-etched silver nanowires (Ag NWs) electrode on stretchable poly-dimethylsiloxane (PDMS) substrate. The pre-charged interface can generate electrostatic interactions with human skin (around eyes) in non-contact mode, and the interacting signal is in accordance with eye movements, which is a different mechanism from previously reported wearable tracker. The pre-etched Ag NWs with serration shape provide a rough surface that can change the inherent capacitance of the interface and the ultrathin dielectric layer is a bilayer structure with polychlorotrifluoroethylene (PCTFE) grafted onto PDMS, which provides interfacial trapping abilities. The triple-layer structure can realize amplified charge storage effect with an ultra-persistent electrostatic charge density in more than thousands of non-contact operation cycles, which allows this AET system to decode oculogyria-induced eyelid movements and related skin fluctuations. Finally, we demonstrate a real-time eye movement and gaze tracking system for visual preference analysis and commercial purpose, as well as an eye-controlled input modality aimed at helping disabled people realize human-computer interactions. The proposed AET system is expected to enrich the approaches of eye tracking techniques and provide a new application of TENG-based electrostatic sensors in medical, commercial and VR areas.

## Results

### Overview of the electrostatic interface and AET system for visual controlled input modality

As shown in Fig. 1a, the electrostatic sensing interface is designed with a triple-layer structure composed of a precharged composite dielectric bilayer and a stretchable back electrode Ag NWs adhered on PDMS substrate. The composite dielectric bilayer is designed by PCTFE grafted onto PDMS. Combined with data acquisition, analysis and progressing parts, an eye-controlled input modality is established (see Fig. 1b) based on the array of the interface arranged in Fig. 1c. Supplementary Fig. 1 is an overview of the working progress of the system and it is similar as the working principle of a non-contact TENG sensor. Based on the coupled effects of 4 channels, the electrostatic signals responding to various eye movements (oculogyria and blinks) can be recognized assisted with deep learning, and the decoded eye movements are then encoded as instructions transmitted to operation systems for realizing eye-controlled mouse actions such as arrow movement as well as clicks or double clicks. Figure 1b depicts the operation progress of eye laevoversion converted into mouse left

movement, which is an application of non-contact sensing and eye-controlled human computer interactions.

The interface is designed to be self-adaptive, flexible and transparent, aimed to be attached on glasses in various curvatures. Figure 1d shows the structure of the composite dielectric bilayer, where the Ag NW layer (the white line) is clearly sandwiched between the substrate and dielectric layer. In this interface, PDMS with oxygen plasma treatment is selected as the flexible substrate owing to its excellent optical transmittance and for better adhesion to the sprayed Ag NW electrode. As shown in Supplementary Fig. 2a, b, the plasma-treated PDMS (PT-PDMS) becomes smoother than the raw film, while a significant adhesion force between PT-PDMS and atom force microscopy (AFM) tip is detected (Supplementary Fig. 2c). The average adhesion force is characterized as 23.21 nN (the insert in Supplementary Fig. 2c) after treatment, which provides notable adhesion to Ag NWs with improved conductivity of the flexible electrode under bending condition (Supplementary Fig. 2d). Ag NWs are etched with serrated bulge (Fig. 1e) for providing a rough surface for the dielectric layer. The inherent capacitance of the interface is optimized by the rough Ag NWs, and a dielectric bilayer film is designed with grafted PCTFE on PDMS which (S-PCTFE) has an improved smooth surface at the nanoscale other than a cracked single layer of sputtered PCTFE (Fig. 1f). The amplified charge storage effect of the rough Ag NWs and composite dielectric layer will be introduced in the next section. The conductivity of the etched Ag NWs after bending and stretching is investigated in Fig. 1g, where the surface resistance is doubled from the initial value of 4.67 Ω when the bending radius is 3 mm. As for the stretching test, when the elongation is 0.1, the resistance experiences a similar increase with the bending test (from 4.38 to 9.31 Ω), and nearly a 3-time increase from 4.72 to 13.46 Ω is observed when stretched to a length of 1.2. The changed resistances of the Ag NW electrode at this scale have almost no influence on the performance of the TENG-based interface due to the high matching resistance of TENG. As shown in Fig. 1h, with the increased layer from single PDMS, the etched Ag NWs (E-Ag NWs) and dielectric bilayer do not lead to an obvious decrease in the visible spectrum area (400–700 nm), and the insert image also exhibits the optical transparency of the interface. In addition, owing to the distribution of the 4 channels around the eye (Supplementary Fig. 3a, b), it will not block the line of sight when tracking eye movements.

### Materials selection and optimization of the electrostatic interface

In TENG-based sensing system, surface charge density can directly determine the sensitivity and the charge-keeping rate is a decisive factor related to the operation stability. As reported before, poly-tetrafluoroethylene (PTFE), polyvinyl chloride (PVC) and PCTFE are commonly used triboelectric polymers, while we carefully studied their charge storage capability for non-contact motion detection. Firstly, the storage charge of the three negative layers is investigated after corona poling. As shown in Fig. 2a, the charges transferred during the first contact are defined as total charges injected on the surface by corona poling. PCTFE shows the highest value (665.75 nC) compared to PVC (524.63 nC) and PTFE (207.76 nC), and the average surface charge density of PCTFE is 1642.46 μC·m$^{-2}$ (Fig. 2b), which greatly exceeds those of PVC and PTFE. These results prove that the polarity of the three materials under corona poling ranks as PCTFE > PVC > PTFE. The dielectric properties of the three materials are also investigated in Supplementary Fig. 4d. To be specific, the surface charge density is not only depended on dielectric property. As for PVC, since its volume resistivity is obviously lower than PCTFE and PTFE (Supplementary Table 1), its weak insulation performance cannot maintain high density of electrostatic charges. It is important to note that the relaxation of electrostatic charge is closely related to the insulation performance of the materials. In this case, the charge-keeping ability is tested by the

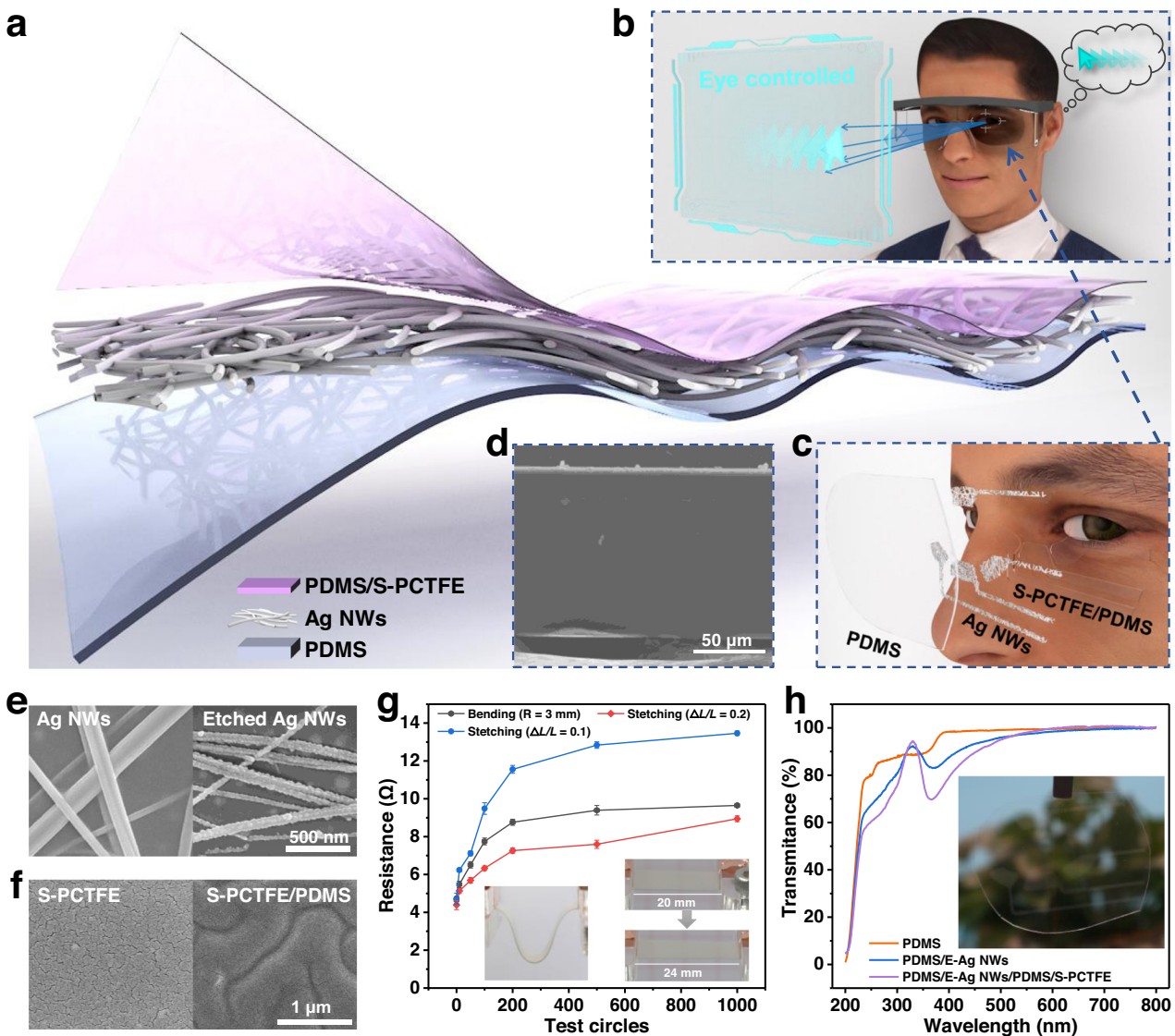

**Fig. 1 | Schematic illustration of TENG-based electrostatic interface. a** Explosive view of the interface. **b** Illustration of the interface-based eye-controlled input modality. **c** The interface array exploded show in front of eye. **d** SEM image of the layered interface. **e** SEM images of the raw and etched Ag NWs. **f** SEM images of the sputtered single layer PCTFE (S-PCTFE) and S-PCTFE grafted on PDMS (S-PCTFE/ PDMS). **g** Surface resistance of bended and stretched Ag NWs electrode on PDMS. (Bending radius at about 3 mm. Data are presented as mean (SD) with error of repeated tests: *n* = 5). Insert: Digital photos of the bended and stretched electrode. **h** UV-Vis transmittance spectra of various layers of the interface. Insert: Digital photo of the interface array.

decay of the open-circuit voltage under non-contact mode TENG (the insert in Fig. 2c). As shown in Supplementary Fig. 4a–c, the trend of the initial voltage is in accordance with that of the transferred charge, where PCTFE shows the highest (24.00 V) among the three films. After 1000 cycles of closing-separation, the voltage of PTFE decays slightly, while that of PVC decreases rapidly in the end. The related charge-keeping rates are shown in Fig. 2c, which illustrates that electrostatic charges on PTFE can remain 96.25% higher than PCTFE (88.65%) and PVC (74.85%). Moreover, diverse charge density and charge-keeping abilities are revealed by density functional theory (DFT) calculations. The electrostatic potential surfaces (EPS) of 10 repeat units of the three materials are shown in Fig. 2d, where PTFE shows a higher potential than PVC, which means that the molecular chain of PTFE is able to withdraw electrons more tightly than PVC, proving the better charge-keeping ability of PTFE than PVC. Conversely, the calculated polarizability summarized in Supplementary Table 1 shows that the average polarization degree of PVC (154.23) is higher than that of PTFE (111.65), which leads to the higher charge density of PVC than PTFE under

corona poling. These results are in accordance with the polarizability of the functional group: **C–Cl** (17.54) > **C–F** (9.16) (Supplementary Table 2). Therefore, the differences in charge density and charge-keeping ability existing among PTFE and PVC are confirmed. The batter charge-keep ability of PTFE is attributed to the higher atom electronegativity of **F** on the molecular chain, while PVC has a higher charge density owing to the higher polarizability of **C–Cl** because the electrons on atom **Cl** are more likely to be delocalized under corona poling. As for PCTFE, since there are both **C–F** and **C–Cl** groups on the molecular chain, the coupled effects contribute to the optimized balance between charge density and charge-keeping ability. As PCTFE shows the highest charge density among the three materials, its voltage decay after 1000 test cycles is still much higher than that of PTFE. Considering the optical transparence of PCTFE is better than that of PTFE (Supplementary Fig. 5), PCTFE is selected as the dielectric layer in this interface.

To realize a stretchable dielectric layer, the thickness of PCTFE is reduced to the nanoscale by magnetron sputtering. However, a single

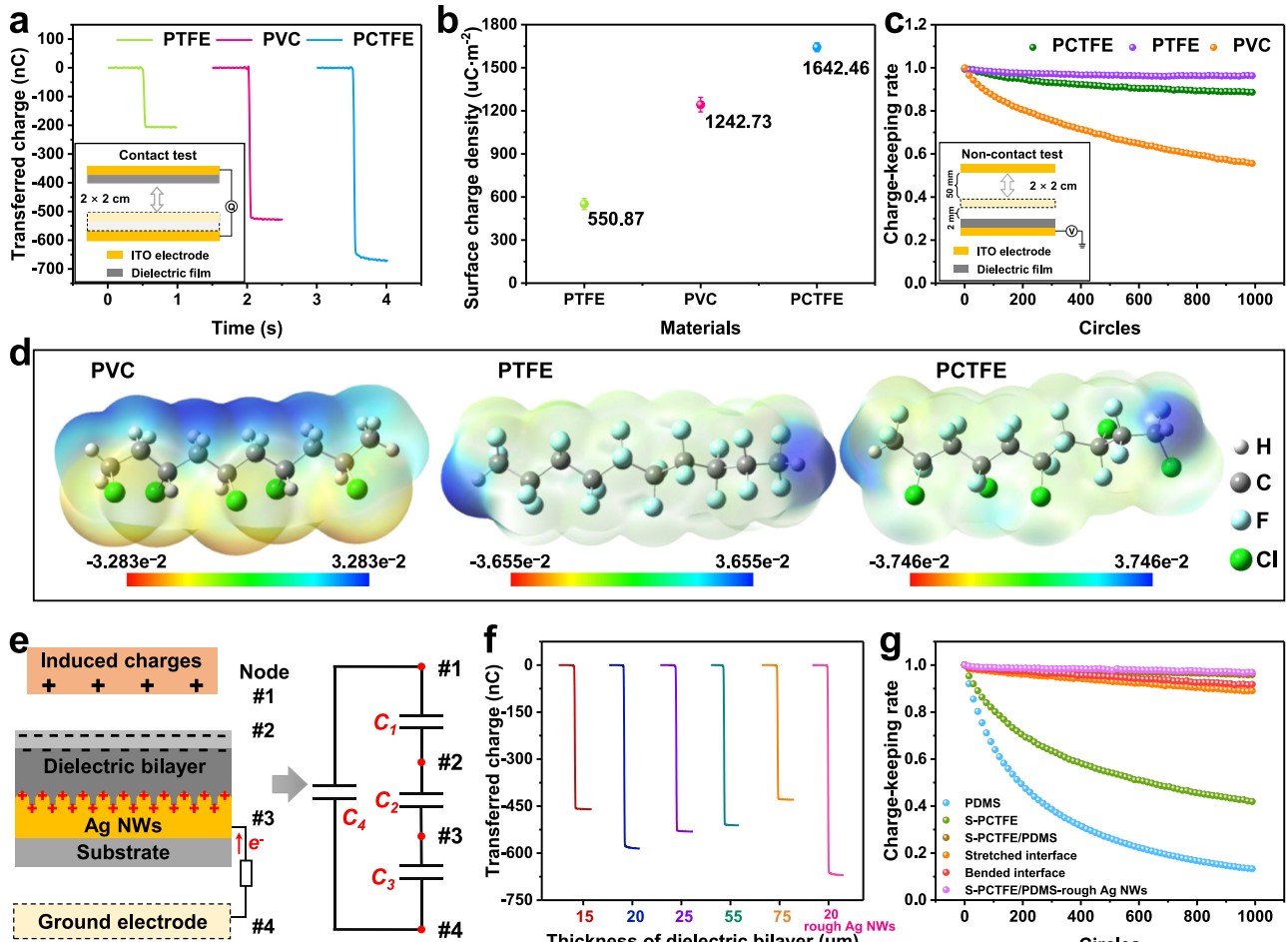

**Fig. 2 | Material-dependent optimization of the interface.** Transferred charge (**a**), average surface charge density (**b**) and charge-keeping rate (**c**) of PTFE, PVC and PTFE films. Insert: Contact and non-contact test method. Data are presented as mean (SD) with error of repeated tests: $n = 5$. **d** Molecular electrostatic potential surface of PVC, PTFE and PCTFE illustrated by 10 repeat units. **e** Layer model of the interface and equivalent circuit. Transferred charge (**f**) and charge-keeping rate (**g**) of dielectric bilayer with various thickness at 15, 20, 25, 55, 75 μm, and rough surface modified interface at raw, bended and stretched state.

nanolayer sputtered PCTFE (S-PCTFE) failed to obtain high transferred charges (17.80 nC in Supplementary Fig. 6a), and the surface charge density of S-PCTFE is only 44.72 μC·m$^{-2}$ (Supplementary Fig. 6b). The initial voltage of S-PCTFE is only 4.31 V and decays to approximately half after repeated testing (Supplementary Fig. 6c). The scanning electron microscopy (SEM) image (Supplementary Fig. 6d) of S-PCTFE illustrated that the depth of the crack in the nanolayer is comparable with the thickness of S-PCTFE (approximately 200 nm measured by a step meter, Supplementary Fig. 6e), resulting in dielectric breakdown across S-PCTFE and poor charge-keeping ability. Meanwhile, the single PDMS layer (30 μm in Supplementary Fig. 6f) is not an alternative choice. As shown in Supplementary Fig. 6a, b, although its transferred charge (61.11 nC), charge density (151.96 μC·m$^{-2}$) and initial voltage (8.96 V, Supplementary Fig. 6g) are higher than S-PCTFE, PDMS cannot maintain electrostatic charges over 13.10% during 1000 non-contact circles because there are no high electronegative groups on its molecular chains.

In this case, a composite dielectric bilayer is proposed by grafting S-PCTFE onto PDMS, and a rough surface Ag NW electrode method is adopted for achieving an amplified charge storage effect. Figure 2e shows a schematic illustration of the modified interface with layered structure and the simulated non-contact electrostatic induction is also summarized in Supplementary Fig. 7. When a charged body (or a grounded body with electrostatic-induced charges) is close to the dielectric layer (precharged by corona poling), the potential difference between back electrode

and the ground appears, driving electrons flowing through the external circuit. Taking the charged body, up surface of the dielectric bilayer and two electrodes as nodes 1–4, an equivalent circuit with four capacitances is established, where $C_1$–$C_3$ are in series and $C_4$ is parallel to them. Here, $C_3$ is the parasitic capacitance of TENG, $C_1$ and $C_2$ can influence the output voltage when operating, and only $C_2$ is controllable to optimize the charge density and charge-keeping ability. For improving the crack surface of S-PCTFE, PDMS is treated by benzophenone with free radicals[42,43], and S-PCTFE can be grafted onto its surface through free radical polymerization by radio frequency (RF) magnetron sputter (Supplementary Fig. 8a)[44–46]. As shown in Supplementary Fig. 8b, the crack surface of S-PCTFE is obviously improved on benzophenone treated PDMS, and the unsaturated group (1650–1750, stretching band of $C = C$ or $C = O$) shown in the Fourier transform infrared (ATR-FTIR) spectra (Supplementary Fig. 8c) is also an evidence of the grafted polymerization. The layered structure of the interface is characterized by energy dispersive spectroscopy (EDS), as shown in Supplementary Fig. 8d. The ultrathin S-PCTFE is revealed by a thin layer in the EDS image of **F** element. The transferred charges of the bilayer are investigated to estimate the influences of thickness of PDMS from 15 to 75 μm in Supplementary Fig. 8e. As shown in Fig. 2f, various transferred charges of the bilayer remarkably exceed those of single S-PCTFE or PDMS in Supplementary Fig. 6a and the highest transferred charges (581.55 nC) are realized with the thickness of 20 μm, where the average charge density reaches

1454.61 μC·m$^{-2}$ (Supplementary Fig. 9a). The thickness of PCTFE is controlled with the sputter time (about 50 nm/h). The transferred charge of the dielectric bilayer with various thickness of PCTFE is investigated in Supplementary Fig. 9b and when the sputtering time is above 2 h, it shows slight influence to the output performance Supplementary Fig. 9c. The charge-keeping ability of the modified interface is also studied in Supplementary Fig. 9d when the thickness of the bilayer layer is 20 μm, its initial voltage is 22.28 V and a high charge-keeping ability (95.92%) is realized, of which both outnumber substantially more than those of single S-PCTFE or PDMS layers. Moreover, the charge-keeping ability over long term are investigated in Supplementary Fig. 9e–g. The initial volage within 15 days decreases gradually to 18.94 V which is about 85% of that of the first day (Supplementary Fig. 9h), while the charge-keeping rate increases slightly to 99.66% after 15 days. The voltage can recover to the initial quantities through recharging progress and multi-recharging initial voltage and charge-keeping rate are shown in Supplementary Fig. 9i. Based on the composite dielectric bilayer, we further adopt rough-surface Ag NWs to modify the capacitance of $C_2$. Through COMSOL calculation, we simplified the dielectric bilayer as a single layer PDMS since the thickness of PCTFE impacts slightly to its output performance (Supplementary Fig. 9c). When the rough dielectric layer is set to the same average thickness as the smooth surface, the capacitance is increased from 426.71 nF to 472.63 nF (Supplementary Fig. 10a, b). Accordingly, Ag NWs are pre-etched by hydrogen peroxide hydrogen peroxide ($H_2O_2$) with serrated bulge, and when PDMS of the bilayer is placed on Ag NWs, it will be cured on the rough Ag NWs and generate a tight layered structure. The final layered structure of the interface is PDMS/Ag NWs/PDMS/S-PCTFE, while rough Ag NWs owns enhanced transferred charges (667.78 nC Fig. 2f), average charge density at (1671.10 μC·m$^{-2}$ Supplementary Fig. 9a) and charge-keeping ability over 96.91% (Fig. 2g) with an initial voltage of 24.39 V (Supplementary Fig. 10c). Furthermore, output signal after bending and stretching are investigated. As shown in Supplementary Fig. 10d, e, although both the stretched and bent interfaces exhibit a decreased initial voltage at approximately 20 V, the stability remains at approximately 90% (Fig. 2g) in the end.

The stabilities of this interface under various humidity, air flow and temperature are also investigated, and the experimental devices are shown in Supplementary Fig. 11a. The influences of humidity are shown in Supplementary Fig. 11b, where the voltage shows slightly decrease until the humidity increased to nearly 90%. As for the air flow, it imposes non-interferences on the output performance of the interface. On the other hand, phenomenon during the temperature experiments shows some interests that are worthy to be analyzed. The voltage signal keeps stable under 40 °C, but increases (from 22.34 to about 51.95 V) within the heating progress from 44.2 to 100.2 °C. Moreover, the voltage signal return to 27.00 V, when the temperature decreases to 27 °C. It has been found that the negative-charged interface can attract positive ions (or particles) in the air[47]. Under the hot air flow, these ions may be blow away or excited to the air, which causes the increased output voltage (the negative charge injected in the film is much more stable). Then, the re-attraction of positive ions can happen during cooling progress, leading to the decrease of voltage signal. Theoretically, the increased output performance is beneficial to improve the resolution of the electrostatic interface, which means our device can work more effectively with higher temperature. Meanwhile, considering surrounding environmental temperature for human rarely exceeds 40 °C, the device can maintain stable operation without the influence from temperature. Finally, the available range (including factors of humidity, air flow and temperature) of this interface is summarized in Supplementary Table 3 with bule background.

## Configuration design of the electrostatic interface array

As shown in Supplementary Movie 1, the periocular movement patterns of eyes are investigated to optimize the configuration of the interface array. Indeed, the eyelids are fluctuated with the contraction and relaxation of periocular muscles. For more precisely characterizing the movements, 3D face scan is conducted, and the details are shown in Fig. 3a and Supplementary Fig. 12. The scan model is built when the eyeball is in the center and rotates to up, down, right and left positions, and movement patterns are analyzed from cross sections (I and II) shown in Supplementary Fig. 12a. The vertical rotation patterns are revealed in cross section I (Fig. 3a). The upper eyelid is at the most convex position when the eye is not rotating, and it will move to the inner direction when eyes rotate upward and downward. Meanwhile, the movement patterns of the upper eyelid are more obvious. When the eyeball is at the center area and the upper eyelid is at the middle position, it will move to be convex when supraversion and become concave when infraversion. In terms of the horizontal rotation of the eyeball, the lower eyelid is worthy of attention. As the cross section shows in Supplementary Fig. 12b, the lower eyelid is moving along with the local muscle's contract and relaxation. When the eyes turn right, the right-side muscles are contracting, and the left is relaxing, followed by the concave area on the right side and the convex area on the left side of the lower eyelid. The same movement patterns are also observed when the eyeball rotates left, where the upper eyelid also moves along with the contracting and relaxing muscles.

The electrostatic interface is designed according to the movement patterns of the upper and lower eyelids. The periocular area that moves substantially is expected to induce more sensitive variations in the electrostatic field on the interface. Therefore, the electrostatic interface array should be arranged as close as possible to the substantial-moving area. As depicted in Supplementary Fig. 13a, a 4-unit interface array is designed which is composed of 2 vertical channels (VC1 and VC2) and 2 horizontal channels (HC1 and HC2), where VC1 is at the position closer to the middle of the upper eyelid when the eye is looking forward, and VC2 is at the middle position in front of the lower eyelid. As for HC1 and HC2, they are located at the two sides of VC1, where they can induce maximum variations in the electrostatic signal with the lower eyelid. The pristine dimensions of the electrostatic interface are shown in Supplementary Fig. 13b. The adhesive PDMS substrate provides self-adhesion to glasses and the flexibility (both electrode and dielectric bilayer) allows the interface to be adhered on glasses with various curvature (Supplementary Fig. 13c), which is convenient to be adopted to diverse glasses such as protective glasses, sun glasses, myopia glasses of different degrees as well as some flexible glasses. The interface can also be stretched to different size (the distance between four electrodes) for adapting various individuals (Supplementary Fig. 13d, the electrode distance is extended with 4 mm).

The array of electrostatic interface can provide a sensitive response to not only eyeball rotation but also blinks. Taking a blink process as an example, the working mechanism is illustrated in Supplementary Fig. 14a, and the electrostatic signals of the 4 channels are shown in Fig. 3b. Before wearing the interface-equipped glasses, the eyelids are wiped with an alcohol pad to eliminate initial surface charges, and human skin is considered as grounded in this kind of electrostatic experiments. At stage (I), the eye is looking forward without any rotation. The bilayer is precharged by corona poling with negative potential, and once a participant wears on glasses equipped with the electrostatic interface, the electrostatic field are induced between interface and eyelashes. The electrostatic signals of the 4 channels at this moment all set to be 0 V. In stage II, when the upper eyelid starts to move down and gets close to VC2, the leaving eyelashes lead to a negative voltage of VC1 and positive of VC2. The eye is totally closed in stage III when the eyelashes are leaving both VC1 and VC2, which induce the lowest potentials of the two channels. When the

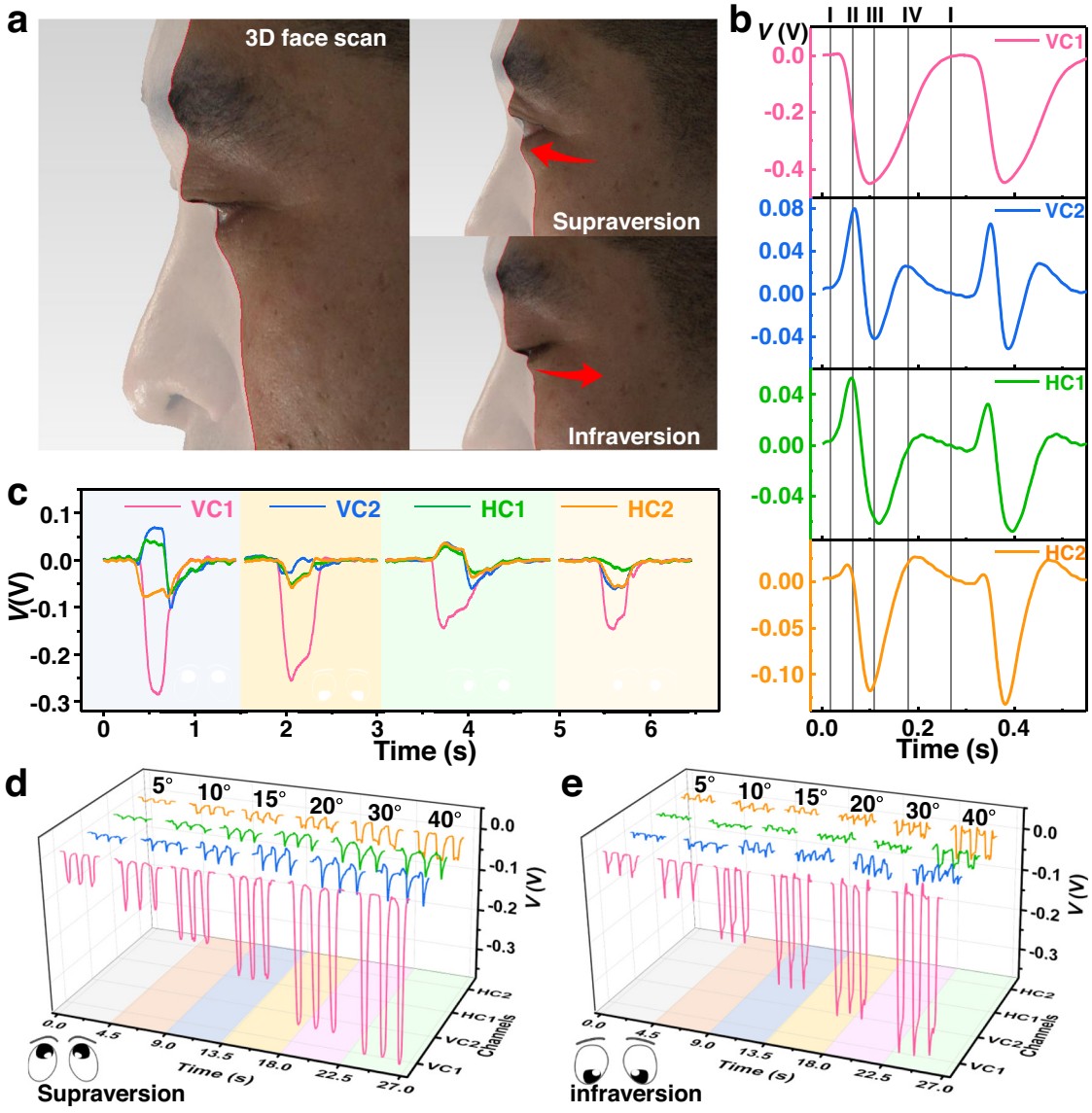

**Fig. 3 | Eye movement patterns and related improvements of the interface array. a** 3D face scan image of supraversion and infraversion. **b** Electrostatic signals of the interface array (including two vertical channels (VC1, VC2) and two horizontal channels (HC1, HC2)) responding to eye blinks. **c** Electrostatic signals of the interface responding to eye movements including supraversion, infraversion, laevoversion and dextroversion). Electrostatic signals of supraversion (**d**) and infraversion (**e**) at degrees from 5° to 40°.

upper eyelid returns to the position at stage IV, another positive potential and medium negative potential are observed in VC2 and VC1. Finally, the eye returns to the initial stage, and the electrostatic signals will also return to the initial value. Specifically, the number of electrons labeled in the figure is just qualitatively describe the direction of the potential. Since HC1 and HC2 are located at the lower eyelid, their signals are the nearly the same trend as that of VC2 (Fig. 3b), which undergoes a slight increase and a plunge in the closing progress and reverse fluctuations when the eye opens. Additionally, the arranged interface array can also provide reliable responses to the rotating eye ball, of which a basic working mechanism is analyzed in Supplementary Fig. 14b and Note 1. As shown in Fig. 3c, distinguishable electrostatic signals are collected when eye movements are supraversion, infraversion, laevoversion and dextroversion (about 30°). As a result of the amplified charge storage effect, the electrostatic field is very sensitive even to slight eyeball rotations. The test method is shown in Supplementary Fig. 15a, b. As depicted in Fig. 3d, e, when the eyeball rotates to

various degrees (supraversion and infraversion from 5° to 40°), the electrostatic signals increase gradually with increasing rotation degree. The same tendencies are also observed when eyes move horizontally (Supplementary Fig. 15c, d), where dextroversion and laevoversion are easy to be identified and the various degrees are tracked observably, which demonstrates the high sensitivity of the interface to small-degree eye movements. Meanwhile, its ability to tracking closing eyes are also demonstrated. As is shown in Supplementary Movie 2 and Supplementary Fig. 15e, the interface can also decode various oculogyric detections with the coupled signals of the four channels, demonstrating the capability applied in monitoring rapid eye movement (REM) sleep. In addition, the eye-tracking performance of the interface are also studied under sweating condition. As shown in Supplementary Fig. 16, the periocular humidity increases from 42.7% to about 71.0% after sweating, which is a humidity range that shows no influence to the stability of the interface (Supplementary Table 3). In this case, the blink (Supplementary Fig. 16b, c) and supraversion

(Supplementary Fig. 16d, e) signals are collected, which both show quite stable after sweating, indicating the stability of this eye-tracking system.

## Eye tracking system applied for preference analysis and eye-controlled input modality

The proposed electrostatic interface array has been proven to have high sensitivity to decode eye movement patterns owing to its ultra-persistent high surface charge density and optimized configuration. Even small-angle eyeball rotations can be tracked, which contributes to an AET system for eye tracking and gaze monitoring. As shown in Fig. 4a, various fruits are set as 3 × 3 in a grid screen and numbered from 1 to 9. When the gaze point moves from one grid to another, the gaze path is recorded by the electrostatic interface as an electrical signal, decoded into movement directions through deep learning, and

finally output as the motion path in the grid screen. The framework system is shown in Fig. 4b containing the grid screen, real-time camera for eye monitoring and electrical output from the interface array. In this system, the vertical and horizontal eye movements are disassembled into 4 directions: upward, downward, right and left. Additionally, diagonal movements are also collected as upper right, upper left, lower right and lower left. The related electrical signals (open-circuit voltage) responding to the various types of eye movements are shown in Supplementary Figs. 17–24 and have been arranged according to gaze point moves in the grid screen. Based on the sensitive interface to eye movements, the movements can be correctly decoded assisted by deep learning through a VGG neural network, as shown in Supplementary Fig. 25a, and the total recognition accuracy can be 97% with low training loss after 40 rounds of training (Supplementary Fig. 25b, c), indicating a high positive predictive value and a true

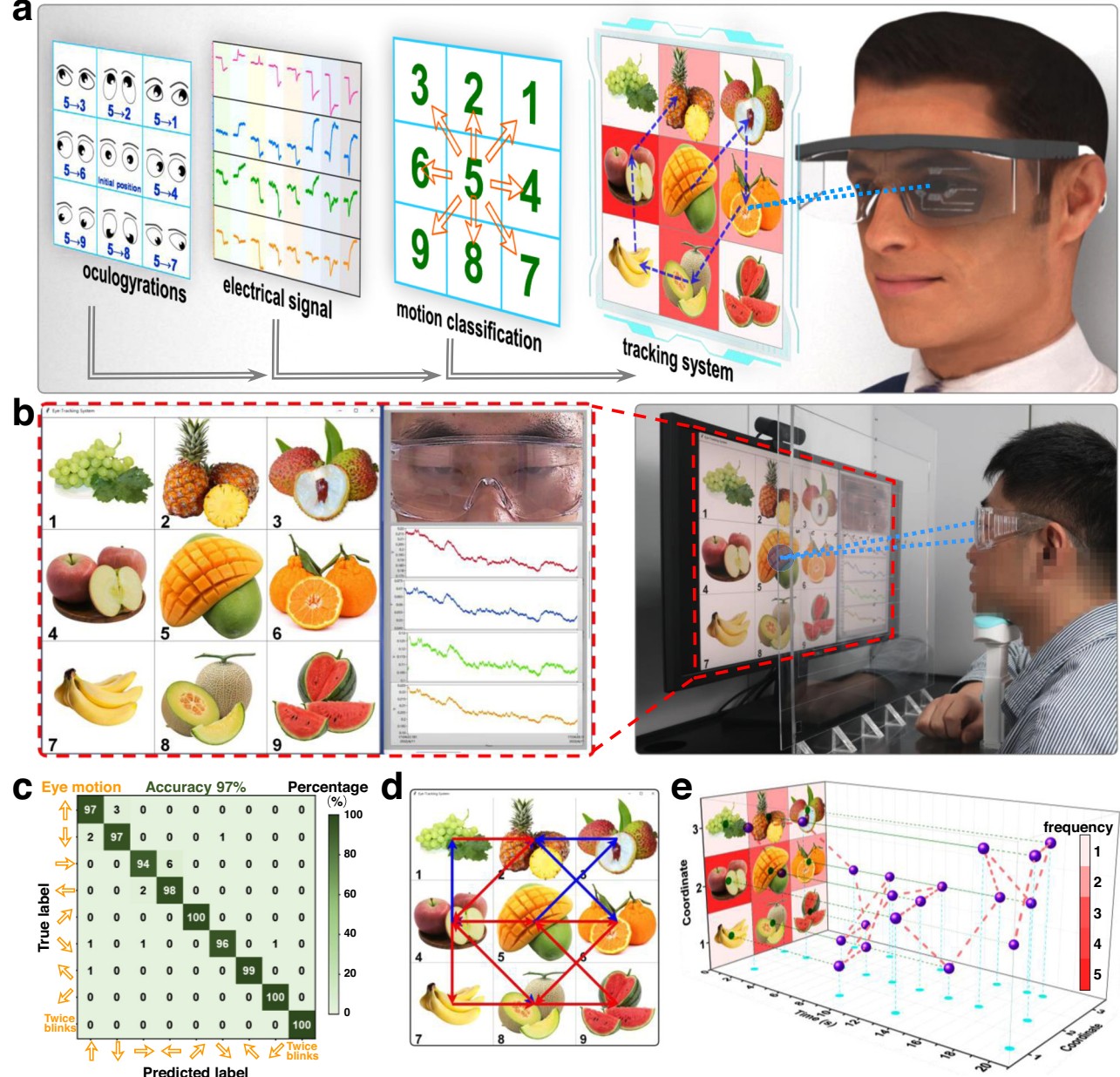

**Fig. 4 | A real-time eye tracking system applied for preference analysis.** Operation progress (**a**) and the working framework (**b**) of the eye tracking system. **c** Confusion map of deep learning outcomes. **d** A real-time "gaze plot" collected from a participant. **e** 3D preference mapping of the participant.

positive rate of eye movement decoding. The total accuracy of the process is 98% (Fig. 4c). Supplementary Movie 3 summarizes the whole tracking progress, and the main system setup is provided in Supplementary Note 2. The eye movements are clearly tracked when the eyesight moves on the grid screen (reverse paths are shown with different colors), and they can be depicted as a moving path in Fig. 4d. Moreover, the moving path can be spread out along with moving times as the 3D mapping (Fig. 4e), where the gaze durations are shown as well as the frequency on each grid as color mapping images. As the eye movement patterns are decoded in real time by the AET system, it is considered to be applied in commercial purpose for collecting areas of interest and visual preference analysis.

Furthermore, an eye-controlled input modality, based on its responses to eyeball rotations and blinks, is designed as a controllable system (Fig. 5a) for people suffering from ALS and Supplementary Note 3 provides the basic setup of the system. In this system, eye movement-induced electrostatic signals (open-circuit voltage) are decoded and encoded into various instructions that can control the mouse actions aimed at users to administer an operation system. Definition and classification of the instructions are supported by deep learning based on the neural network module shown in Fig. 5b. Since eye movements can be accurately recognized by electrical signals, a simple neural network containing three superposed Conv structures of Conv1d+Relu+Bn and a fully connected layer is able to decode the movement patterns. In the training progress, a database of the electrostatic signals responding to 8 types of eye movements (2 blinks, 3 blinks, 4 blinks, double 2-second gaze, supraversion, infraversion, dextroversion and laevoversion) is set and encoded as 8 instructions to control the mouse input action as left click, right click, double clicks and dormant (Fig. 5c), as well as the mouse movement toward up, down, right and left directions, respectively (Fig. 5d). The 8 signals are visibly distinguishable, and after the data progress of the VGG classification, the test accuracy reaches 100% after 20 eopches (Supplementary Fig. 26a). Meanwhile, a low train loss is also obtained in Supplementary Fig. 26b and the total accuracy of the process is 100% (Supplementary Fig. 26c). Then, the real-time recognition and operation progress are conducted, when wearing glasses assembled with the interface, the user can only control actions of the mouse with his eyes. In Fig. 5e and Supplementary Movie 4, the eye-controlled file copying process is depicted, which not only shows the real-time electrical signals and operation windows but also records the eye movements. In this process, eye movements are precisely transformed into mouse actions simultaneously with a negligible time interval for data processing (within 1.002 ms, Supplementary Fig. 26d), including directional movements, left click (file selection), double click (file opening) and right click (menu taking), which is a handless eye-controlled input modality for ALS patients in human-computer interaction areas. Since this eye-tracking system is also sensitive to closed-eye movement, it has the potentiality to monitor REM sleeps. Furthermore, the eye tracking for patients with vegetative state (VS) is a future application since eye movement is important clinical features. This active electrostatic interface equipped with soft glasses is a potential approach for clinical diagnosis of VS patients (Supplementary Fig. 27a). Besides, as the interface is based on the electrostatic induction with skin fluctuation, it can also be used for other skin-based sensing, such as monitoring the movement of larynx (Supplementary Fig. 27b), which is related with swallowing diseases. Except body sensing, this electrostatic can also realize contactless human-machine interactions to avoid direct contact in some cleaning environment (Supplementary Fig. 27c).

## Discussion

A non-contact eye-tracking system has been proposed based on a flexible, transparent and highly persistent electrostatic interface, which can generate output voltage signal based on the electrostatic induction between the charged interface and periocular skin. A

stretchable bilayer with a thickness of 20 μm is fabricated by grating PCTFE on PDMS and its crack surface is improved since PDMS is treated with free radicals for the grafted PCTFE. Moreover, Ag NWs etched with a rough surface in nano scale is applied as the electrode, where the induced serration surface of Ag NWs can increase the inherent capacitance, leading to the enhanced charge maintenance. The amplified charge storage effect of the triple layer (dielectric bilayer and rough electrode) realizes a high surface density of 1671.10 μC·m$^{-2}$ and the persistence of electrostatic charge reaches 96.91% after 1000 non-contact operation cycles. The output performance of the interface is stable at humidity under 90% and air flow of 8.1 m/s, while the output voltage can be increased with the increase of temperature, which is an interesting phenomenon. This effect can achieve an ultrasensitive non-contact sensory interface for tracking eyeball movements and related skin fluctuations, based on simple electrostatic induction effect. The configuration of the interface array is investigated through analyzing fluctuations of the periocular skin by 3D face scan, and the optimized array is arranged as two vertical channels above and below the dialect to the eye ball and another two horizontal channels located at either side of the lower eyelid. Accordingly, the interface array can detect the eye movements of blinks and oculogyria with an angular resolution of 5°, and it is also capable of decoding oculogyric directions of closing eyes. Finally, an AET system is established based on deep learning process. We demonstrate the application of the system to real-time decode eye movements and gaze replays for analyzing commercial preferences. Then, an eye-controlled human-computer interaction is demonstrated as an approach to emancipate hands or to help ALS patients. The AET system is considered as a promising access that can be used in a wide range of applications in commercial and engineering but not limited in monitoring REM sleep as well as fatigue-driving warnings. Supplementary Table 4 summarizes the advantages and disadvantages of electrical trackers compared with other well-known methods. After that, the EOG and TENG-based tracker, as the most typical non-invasive electrical method, are compared in Supplementary Table 5. The glasses-integrated sensing interface and its non-contact tracking mechanism endow this AET system with enhanced portability and wearability, while EOG sensor may bring discomfort feeling due to skin breath and infectious risk. Meanwhile, the simple structure, non-contact working mode and closing-eye tracking ability make this TENG-based tracker worthy promoting.

## Methods
### Materials
Polymer films including PTFE, PVC, and PCTFE with thicknesses of 20 μm are available on the market without further modification. H$_2$O$_2$, poly(vinylpyrrolidone) (PVP, average MW = 60000) benzophenone, PDMS, and absolute ethanol were purchased from commercial sources without further purification.

### Plasma treated PDMS substrate
PDMS (SYLGARD184 Dow Corning) main agents and curing agents were mixed thoroughly by weight ratio of 15:1 and degassed under vacuum for 5 min to remove air bubbles. The degassed mixture (0.3 ml) was spin-coated in a plastic dash (radius = 30 mm) and cured at 80 °C for 4 h. Thus, the cured film substrate film was controlled at approximately 100 μm. The film was finally treated with O$_2$ (80 sccm gas flow) plasma to obtain a plasma-treated PDMS film with enhanced adhesion force.

### The etching of Ag NWs
Ag NWs (100 μL, 20 mg·mL$^{-1}$ in absolute ethanol, XFNANO) was added into a solution of PVP (10 mL, 1 mg·mL$^{-1}$, deionized water) in an ice bath to obtain a dispersed mixture. Then, after 333 μL H$_2$O$_2$ (30%) was added, the mixture was vibrated for 1 min and then centrifuged at

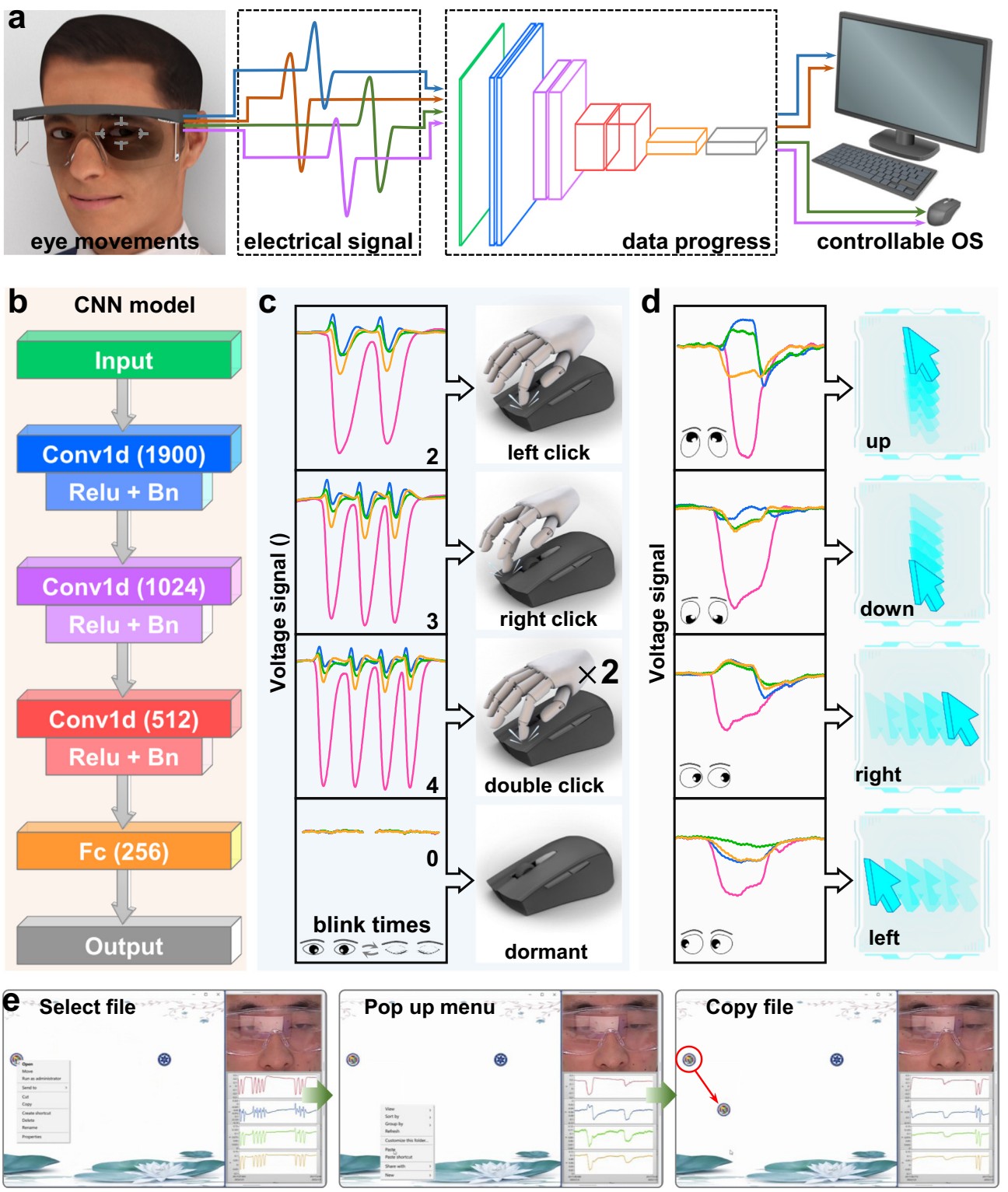

**Fig. 5 | Demonstration of an eye-controlled input modality. a** Operation framework of the modality. **b** Neural network module of deep learning utilized in the system. **c**, **d** Decoded eye movements and consistent encoded input actions of the system. **e** A file-copying progress enabled by the system.

6790 g for 10 min. The Ag NWs were washed with ethanol (25 mL), centrifuged repeatedly 3 times and finally dispersed in ethanol (2 mL) to obtain the etched Ag NWs (approximately 1 mg·mL$^{-1}$).

**Fabrication of the interface**
Etched Ag NWs (1 mg·mL$^{-1}$) were sprayed for 3 s with a mask (hollowed location of the electrode array) on the plasma-treated PDMS substrate

and dried at 70 °C for 1 min. The spraying and drying were repeated 20 times to obtain the Ag NW electrode array. Then, another PDMS layer (mixture of main and curing agents, mixing ratio of 15:1 by weight) was spin coated to overlay Ag NWs and cured at 80 °C for 4 h. To obtain bilayers with various controlled thicknesses (75, 55, 25, 20, 15 μm), the spinning speed was set at 1, 2, 3, 4, 6 krpm. The new cured PDMS layer was immersed with a solution of benzophenone (10 wt% in ethanol)

and exposed to ultraviolet light (365 nm) for 1 h. After that, the treated surface was washed with ethanol and completely dried with nitrogen gas 3 times. Finally, with a mask on the new-treated benzophenone-treated PDMS, SPCTFE was sputtered (Ar 60 sccm, 60 W, about 50 nm/h) on the position overlapped with the electrode, and the layered film was cut out to obtain the interface array.

### Stability test under environmental factors
The experimental devices are shown in Supplementary Fig. 12a and the effective area of the interface is $2 \times 2$ cm with the closing-seperation circle at 2–50 mm (the insert in Supplementary Fig. 11a). The humidifying, air flow and heating are started at 30 s, with the factors (humidity, speed of air flow and temperature) recorded every 30 s.

### Human experiments
5 volunteers (including 2 male and 3 female, aged from 20 to 40 years old) are recruited to help execute the AET system. The periocular humidity is measured by put the probe near the outside of the eye before and after sweating with one of a male volunteers. Our research complies with all relevant ethical regulations, overseen by Committee on Ethics of Beijing Institute of Nanoenergy and Nanosystems (A-2019027). All data measurements on bodies of participants are performed with their full, informed consent.

### Execution of the AET system
As shown in Fig. 3b and Supplementary Fig. 13a, the interface array was attached on the glasses with its geometrical center coincided with the pupil of the wearer, and 4 channels are wired with the multichannel data acquisition device. Before wearing, the eyelids of wear are wiped with an alcohol pad to eliminate initial surface charge, and human skin is supposed to be grounded. when adhering the interface, a center in in front of pupil (eyes looking forward) on the glasses should be pre-mask to calibrate the center of the interface. Then the glasses should be wore appropriately making the interface at the right position. Before activating the decoding system, 2 blinks action and the related signal is induced to approximately execute the amplitude of VH1 (about 0.45 V) from the signal window (the upper right window in Supplementary Movie 2, 3). After that, the real-time tracking system is activated then enter running state.

### Characterization and measurements
A programmable electrometer (Keithley 6514) was applied to test the open-circuit voltage and transferred charge. An SEM (SU8020, Hitachi) equipped with EDS accessory was used to characterize the morphologies of Ag NWs, thickness of the bilayer and layered structure of the interface. A UV-Vis-NIR spectrophotometer (UV-3600, Shimadzu) was used to measure the UV–Vis transmission spectra of the interface and various films. A source meter (Keithley 2450) with a four-point probe (HP-504, 4Probes Tech Ltd.) was used to characterize the surface resistance of Ag NWs. AFM (MFP-3D from Asylum Research) was used to analyze the surface morphology and force curve of PDMS. A 3D face scanning system (FC BodyScan, Weinan Lingzhi 3D Technology) was applied to obtain the three-dimensional geometry of eye movements. A broadband dielectric/impedance spectrometer (Novocontrol Concept 90) was used to measure the dielectric permittivity spectra. A multi-channel data acquisition device (NI USB-6356, National Instruments) assisted with LabVIEW (2021) was used to collect the electrostatic signal of the interface array. A plasma system (IoN 40, PVA TePla) was used to treat PDMS with improved adhesion force. A sputter deposition platform (Discovery 635, DentonVacuum) was used to deposit the PCTFE nanofilm. A step meter (Dektak XT, Bruker) was used to measure the thickness of the sputtered PCTFE.

### Statistics and reproducibility
Each experiment (including SEM images) was repeated independently with similar results.

### Reporting summary
Further information on research design is available in the Nature Portfolio Reporting Summary linked to this article.

## Data availability
The authors declare that all data supporting the results in this study are present in the paper and the data sources are uploaded in Supplementary Information with this paper. Source data are provided with this paper.

## Code availability
The codes that support the findings of this study are available from the corresponding authors by request.

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

## Acknowledgements

We thank Doctor Shanshan Du (Peking University Third Hospital) and Ping Chen (Weinan Lingzhi 3D Technology Co., Ltd.) for providing the 3D face scan techniques. This work was supported by the National Key R&D Project from Minister of Science and Technology (2021YFA1201601), National Natural Science Foundation of China (Grant No. 62174014), Beijing Nova program (Z201100006820063), Youth Innovation Promotion Association CAS (2021165), Innovation Project of Ocean Science and Technology (22-3-3-hygg-18-hy), State Key Laboratory of New Ceramic and Fine Processing Tsinghua University (KFZD202202), Fundamental Research Funds for the Central Universities (292022000337), Young Top-Notch Talents Program of Beijing Excellent Talents Funding (2017000021223ZK03).

## Author contributions

Y.S., X. Chen. and Z.L.W. conceived the idea and designed the project. Y.S. prepared the manuscript. X. Chen., X. Chu., and Z.L.W. revised the manuscript. Y.S. and P.Y. designed the structure of the device. Y.S. and R.L. performed the data measurements and analysis. Z.L. and X.D. assisted in the fabrication and characterization. X.T. worked on AFM measurements. All the authors discussed the results and commented on the manuscript.

## Competing interests
The authors declare no competing interests.
