## [Peer review file · Nature Communications]

REVIEWER COMMENTS

Reviewer #1 (Remarks to the Author):

In this manuscript, the authors demonstrated a transparent, flexible and ultra-persistent electrostatic sensing interface for realizing active eye tracking (AET) system based on the electrostatic induction effect. The authors prepared a triple-layer structured composite consisting of a dielectric bilayer (PCTFE grafted onto PDMS) and a rough-surface Ag nanowire electrode layer, leading to the enhancement of capacitance and interfacial trapping density, which improves the triboelectric performance. Furthermore, the authors designed an electrostatic interface with 4 channels (2 vertical and 2 horizontal ones) according to the periocular movement patterns of the eyes. Through the deep learning process, the authors showed a real-time eye-tracking system applied for preference analysis and eye-controlled input modality. While the authors showed a practical application of the EOG sensor based on electrostatic induction, the scientific explanation of the mechanism of the non-contact EOG sensor is not clear and the material characteristics are not enough. In addition, previous literatures about EOG sensor are not well introduced and compared in the introduction. This reviewer cannot recommend this manuscript for publication in Nature Communications. Below are additional comments.

1. The authors are recommended to discuss the effect of plasma treatment on the surface roughness and adhesion force. In addition, please provide the evidence and reasonable discussion of grafting PCTFE on PDMS. Why do the surface crack and wrinkles appear in Fig. 1f and Supplementary Fig. 8?
2. The authors need to explain the detailed reason for the higher polarizability of PCTFE than the others. In addition, please provide results and a discussion about the dielectric properties of materials.
3. The thickness of each layer in the bilayer film (S-PCTFE-grafted PDMS) is a critical factor affecting the device's performance. The authors should optimize the film thickness of PCTFE and PDMS and discuss the effect of thickness on the output performance.
4. Among the surface charge density and charge-keeping rate, which one is the more critical factor affecting the device's performance and reliability?
5. In Supplementary Fig. 10a-b, the authors should provide a discussion and experimental results of dielectric bilayers.
6. Authors need to clearly explain the working mechanism of the device that is sensitive to the eyeball motions of the eyes. Although the authors designed the electrostatic interface according to the movement patterns of the eyelid, the authors explained the working mechanism based on the movement of eyelashes (Supplementary Fig. 13) instead of eyeball rotation.
7. In Supplementary Fig. 13, considering electron flows in VC2 at stage II~IV, the electric signals in HC1 and HC2 should be similar to those in VC2. However, different signals and intensities are shown in Fig. 3b
8. The authors should clearly explain output signals in Fig. 3c and Supplementary Fig. 14. Based on the mechanism, the authors should explain the results in Supplementary Fig. 15~22.

9. The authors should introduce previous literatures on electric signal-based eye trackers. In the reference “Imperceptible electrooculography graphene sensor system for human–robot interface” (npj 2D Materials and Applications 2018), the EOG sensor has been developed, which is responsive to the motion of eye blinking. In addition, the authors should introduce previous literatures on EOG sensors based on the triboelectric mechanism.

Reviewer #2 (Remarks to the Author):

This work uses the principle of non-contact electrostatic induction in a contactless triboelectric sensor, using an array of four sensors on the glasses.

The authors use the eyelesh for detection of real-time eye movements and using this concept they created eye-controlled human-computer interactions.

The work is presented well, and results are satisfactory with details implementations for some unique applications. The design of sensor has been done in an optimum way to obtain maximum performance. Through Machine learning they are able to Identify the movement of the eye using classification of the signals of the four sensors.

Comparing to the literature on eye tracking, some of them have been cited as well, I find the contributions is limited for this journal.

It will be interesting to include the electronics used for data acquisition.

Reviewer #3 (Remarks to the Author):

Eye tracking is a method of recording and analyzing eye movements to gain insights into a person's visual attention and thought processes. It is used in various fields, such as virtual reality, human-computer interaction, and medical monitoring, providing valuable information on visual attention, cognitive processes, and behavior. The authors of this manuscript propose a new type of active eye tracking (AET) system, based on the electrostatic induction effect. This system is designed to be transparent, flexible, and ultra-persistent, offering improved capabilities for monitoring and analyzing eye movements. The proposed AET system uses a triple-layer structure to enhance its performance. The structure consists of a dielectric bilayer and a rough-surfaced Ag nanowire electrode layer, which improves the intrinsic capacitance and charge storage capability of the electrostatic interface through

increased interfacial trapping density. This advanced system offers oculogyric detection with an impressive angular resolution of 5° . The results in this study is robust and well-supported by comprehensive set of experimental data. The real-time decoding of eye movements, enabled by the AET systems, can be of great use for many envisioned applications. Below, please find several questions and comments which can help the authors improve the manuscript.

Q1. The authors utilized 4 channels in the interface array. Can the experimental or theoretical basis for determining this number of channels be discussed? Additionally, it would be helpful to have an explanation of how the performance of the system is affected as the number of channels increases or decreases.

Q2. In the field of object recognition, several models such as AlexNet and ResNet exist. Could you explain why VGGNet was selected over other models for this study?

Q3. It is mentioned in line 345 that the AET system has the capability of real-time classification. Can the authors specify the latency and processing time of the system?

Q4. According to the information provided in Supplementary Table 3, the system can detect close eye tracking. Can the system also detect eye movements when the eyes are closed?

Q5. The line 387 mentions that the initial electrostatic signals of the 4 channels were determined through 2 blinks. Can the authors provide more detail on this process? Specifically, what data was used to determine the optimal location for the channels?

Q6. Can this system potentially detect the change of pupil size as well?

Reviewer #1 (Remarks to the Author):

In this manuscript, the authors demonstrated a transparent, flexible and ultra-persistent electrostatic sensing interface for realizing active eye tracking (AET) system based on the electrostatic induction effect. The authors prepared a triple-layer structured composite consisting of a dielectric bilayer (PCTFE grafted onto PDMS) and a rough-surface Ag nanowire electrode layer, leading to the enhancement of capacitance and interfacial trapping density, which improves the triboelectric performance. Furthermore, the authors designed an electrostatic interface with 4 channels (2 vertical and 2 horizontal ones) according to the periocular movement patterns of the eyes. Through the deep learning process, the authors showed a real-time eye-tracking system applied for preference analysis and eye-controlled input modality. While the authors showed a practical application of the EOG sensor based on electrostatic induction, the scientific explanation of the mechanism of the non-contact EOG sensor is not clear and the material characteristics are not enough. In addition, previous literatures about EOG sensor are not well introduced and compared in the introduction. This reviewer cannot recommend this manuscript for publication in Nature Communications. Below are additional comments.

Answer) We thanks for this kind review and we are very sorry that we didn't clearly explain the difference between our device and previous EOG sensor. The mentioned EOG tracker is a promising approach based on detecting the deflection of the corneo-retinal dipole. The EOG sensor can maintain high resolution based on pairs of electrodes placed around eyes and it has been successfully applied in both research and medical diagnosis. However, the non-contact active eye tracking (AET) system is quite different from EOG sensor. The non-contact AET system depends on the electrostatic induction between the static-charged interface and human skin (around the eyes). In our system, static-charged interface is attached on the inner surface of an eyeglass, which is suspended in front of eyes with a distance around 1 cm. Here, the periocular skin in this AET system is considered as grounded (as reported in: *Sci. Adv.* 2022 8, eabo5201; *Adv. Mater. Technol.* 2019, 1900789). Then, the moving skin around the eyes can disturb the electrostatic field generated by static-charged interface, resulting in the potential change as the sensory signal.

It is important to note that the electrostatic interface in our work is pre-charged with maximized charge density and thus, it can generate sensory signal in non-contact mode.

This special characteristic is quite different from previous EOG sensor. In this non-contact AET system, human skin has no direct contact with the sensor electrodes, which is a different working style for eye-tacking. The non-contact tracking mode facilitates the portability and wearability of this AET system, and we believe it is an alternative choice for EOG sensor. The mode innovation of eye-tracking system, such as our non-contact AET system, can promote the study field of medical monitoring and human-computer interactions. From aspect of user experience, this AET system likes a general glasses for myopic correction and EOG sensor is more like a contact lens for good appearance, both of which have their own loyal customers.

We further clarified this part and added introduction about EOG sensor, as can be seen on page 2–3, and updated reference list about EOG sensor as following:

- [22]. Multipurpose and reusable ultrathin electronic tattoos based on PtSe₂ and PtTe₂. *ACS Nano* **15**, 2800-2811 (2021).
- [27]. Fabrication, characterization and applications of graphene electronic tattoos. *Nat. Protoc.* **16**, 2395-2417 (2021).
- [28]. Imperceptible electrooculography graphene sensor system for human–robot interface. *npj 2D Mater. Appl.* **2**, (2018).
- [30]. Highly stretchable starch hydrogel wearable patch for electrooculographic signal detection and human–machine interaction. *Small Struct.* **2**, (2021).

1. authors are recommended to discuss the effect of plasma treatment on the surface roughness and adhesion force. In addition, please provide the evidence and reasonable discussion of grafting PCTFE on PDMS. Why do the surface crack and wrinkles appear in Fig. 1f and Supplementary Fig. 8?

Answer) We thanks for this kind recommendation. As shown in Supplementary Fig. 2 a, the roughness of the plasma-treated PDMS is decreased slightly, which is also revealed with the added SEM image (Supplementary Fig. 2c). Importantly, the average adhesion force increased from 4.95 nN to 23.21 nN after treatment. The increased adhesion ability provides stable conductivity for the flexible electrode, which is investigated through the resistance in Supplementary Fig. 2d, where the treated PDMS-based electrode shows smaller resistance than that of raw PDMS.

Supplementary Fig. 2. Plasma-treated PDMS. (a–c) AFM images for dimensional surface morphology (a), SEM images (b) and force curves with insert adhesion force (c) on PDMS with and without plasma treated. (d) Surface resistance of bended Ag NW electrode on raw and plasma-treated PDMS.

As for the grafted PCTFE on PDMS, the similar treatment process has been well explained before. UV-excited benzophenone is used as the initiator to promote radical grafting or polymerization of various hydrophilic monomers onto PDMS^{42,43}. On the other hand, Radio frequency (RF) sputtering has a long history applied for polymerization, for example PTFE, FEP and PE^{44,45}. In the sputtering process, energetic positive ions bombard the polymeric target surface and break the polymer chains. Volatile fragments are emitted into the plasma volume and create precursors (free radicals) of the plasma polymerization process⁴⁶. Thus, when the substrate (PDMS) is initiated with free radicals, the fragments radicals will bond to generate the grafted dielectric bilayer (Supplementary Fig. 8a). As we reported before, the unsaturated groups on the sputtered PTFE are the evidence for the free radical polymerization⁴⁷. In the ART-FTIR spectra of sputtered PCTFE on PDMS, the unsaturated group also shows obvious peaks ($1650\text{--}1700\text{ cm}^{-1}$, Supplementary Fig. 8b) which are related to the generated C=C or C=O group.

Supplementary Fig. 8. Fabrication and characterization of the dielectric bilayer. (a) Fabrication progress of the bilayer. (b) Nano morphology of S-PCTFE sputtered on raw and benzophenone-treated PDMS. (c) ATR-FTIR spectra of PDMS, PCTFE and the bilayer (PT-PDMS/PCTFE). (d) Energy Dispersive Spectroscopy image (Cross section) of the layered interface (Elements: F, Ag, Si). (e) Cross section of the dielectric bilayer layer with various thickness controlled by spin coating speed.

The sputtered PCTFE (Cu foil as substrate) in Fig. 1f shows crack structure, while it forms a dense film sputtered on benzophenone treated PDMS, and the film are smoother than PCTFE on raw PDMS (Supplementary Fig. 8b). This is also another evidence of the grafted dielectric bilayer. As for the wrinkles, it is the structure that generate in the long-term sputtered progress by the plasma.

We modified the paper to address this comment (page 4 and 7), and updated references as following:

- [42]. The relationship between bulk silicone and benzophenone-initiated hydrogel coating properties. *Polymers* **10**, (2018).
- [43]. Photopatterning of pdms films: Challenging the reaction between benzophenone and silicone functional groups. *Materials* **14**, (2021).
- [44]. RF magnetron sputtering of polymers. *J. Non-Cryst. Solids* **218**, 44-49 (1997).
- [45]. Stelmashuk, V., Biederman, H., Slavínská, D., Trchová, M., Hlidek, P. Rf

magnetron sputtering of polypropylene. *Vacuum* **75**, 207-215 (2004).

[46]. Contributions of different functional groups to contact electrification of polymers. *Adv. Mater.* e2001307 (2020).

2. The authors need to explain the detailed reason for the higher polarizability of PCTFE than the others. In addition, please provide results and a discussion about the dielectric properties of materials.

Answer) Thanks for this suggestion. We have investigated the dielectric spectra of the three materials in the revised manuscript (Supplementary Fig. 4c). As we calculated through DFT (Fig. 2d, Supplementary Table 1 and 2) and discussed in the manuscript, PCTFE has the highest electrostatic potential surfaces and molecular polarizability, because of the higher polarizability C–Cl functional group, which originates from the delocalized effect of Cl under electric field. These characters endow PCTFE the highest initial voltage (Supplementary Fig. 4). As for the dielectric properties, it is investigated in Supplementary Fig. 4d. To be specific, the surface charge density is not only depended on dielectric property. As for PVC, since its volume resistivity is obviously lower than PCTFE and PTFE (Supplementary Table 1), its weak insulation performance cannot maintain high density of electrostatic charges. It is important to note that the relaxation of electrostatic charge is closely related to the insulation performance of the materials. In this case, the charge-keeping ability is tested by the decay of the open-circuit voltage under non-contact mode TENG in Supplementary Fig. 4a–4c. PCTFE shows the highest initial voltage (24.00 V), and the decayed voltage (after 1000 cycle) is still high than the initial voltage of PTFE. As PCTFE shows the highest charge density among the three materials and considering its optical transparency is better than that of PTFE (Supplementary Fig. 5), it is elected as the dielectric layer in this interface. We further clarified this part, as can be seen on page 5.

Supplementary Fig. 4. Output performance and dielectric property of PVC, PTFE and PCTFE. (a–c) The open-circuit voltage of PVC (a), PCTFE (b) and PTFE (c) under non-contact condition. Repeat times: 1000. (d) Dielectric permittivity spectra of the three materials.

Directions	XX	YY	ZZ	Average	Volume resistivity (ohms·cm)
PTFE	137.43	97.91	99.60	111.65	1E+18
PVC	176.17	152.71	133.79	154.23	13E+13
PCTFE	177.63	152.74	144.43	158.27	2E+16

Supplementary Table 1. Calculated molecular polarizability and volume resistivity of PTFE, PVC and PCTFE.

3. The thickness of each layer in the bilayer film (S-PCTFE-grafted PDMS) is a critical factor affecting the device's performance. The authors should optimize the film thickness of PCTFE and PDMS and discuss the effect of thickness on the output performance.

Answer) This is a very kind suggestion. We have added the investigation of the thickness about sputtered PCTFE. The thickness of PCTFE is controlled with the

sputtering time (about 50 nm/h) on the PDMS (thickness: 20 μm). The detailed sputtering condition is listed in the Method part (Ar 60 sccm, 60 W). The transferred charges and surface charge density of these samples are measured in Supplementary Fig. 9b. When the sputtering time is above 2 h, it shows slight influence to the output performance, which is in accordance with the results of the dielectric property. The dielectric constant changes slightly with the thickness of grafted PCTFE, because the thickness of PCTFE (nano scale) is almost neglect compared with the thickness of PDMS (20 μm). More importantly, the triboelectrification is still an interfacial effect. For the PCTFE film with current fabrication method, the space charge in the bulk region is rather insignificant and thus, the thickness effect is also very weak.

Supplementary Fig. 9 (b and c) The output performance (b) and dielectric property (c) of the dielectric bilayer with various sputtering time of PCTFE (depositing thickness 50 nm/h) on PDMS (thickness: 20 μm).

As for the thickness of PDMS, it is discussed in Fig. 2f and Supplementary Fig. 9a. The PDMS in thickness of 20 μm can provide the optimal output performance. Therefore, the optimized thickness of the dielectric bilayer is based on 20 μm PDMS with grafted PCTFE about 200 nm. We further clarified this point, as can be seen on page 7.

4. Among the surface charge density and charge-keeping rate, which one is the more critical factor affecting the device's performance and reliability?

Answer) This is a valuable question. The high surface charge density means the sensitive electrostatic interactions between the charged interface and the periorcular skin. Thus, the PCTFE with highest surface charge density contributes to the high sensitivity of this AET system. Concerning the charge-keeping rate, it is related with the stability of the AET system and it can directly determine the applicability of the device. A persistent high charge-keeping rate facilitates the stable operation. Hence, they are both

the critical factor of the AET system, but the charge-keeping rate should be more important in this case. We further clarified this part, as can be seen on page 5.

5. In Supplementary Fig. 10a-b, the authors should provide a discussion and experimental results of dielectric bilayers.

Answer) Thanks for this kind suggestion. When simulating the capacitance with the rough surface, we induce PDMS as the dielectric layer. As discussed in Supplementary Fig. 9c, the influence of the thickness of PCTFE to the dielectric property of the bilayer is nearly negligible. From this aspect, when simulating the capacitance with the rough surface, the dielectric bilayer is simplified as a single layer PDMS. The good flowability makes PDMS solidified on the pre-etched Ag NWs with rough surface, which modifies the inherent capacitance of the TENG model leading to the enhanced charge-keep ability. We further clarified this part, as can be seen on page 8.

6. Authors need to clearly explain the working mechanism of the device that is sensitive to the eyeball motions of the eyes. Although the authors designed the electrostatic interface according to the movement patterns of the eyelid, the authors explained the working mechanism based on the movement of eyelashes (Supplementary Fig. 13) instead of eyeball rotation.

Answer) We appreciate this kind suggestion. As we investigated in the Fig. 2a, Supplementary Fig. S11 and Movie S1, the motion ranges of the upper eyelid are the most obvious than other parts of periocular skin. Indeed, the patterns of vertical rotation of eyeballs are similar to that of blink, which is the reason for the similar signals between blink and vertical oculogyria. However, the motion of the eye is a cooperative pattern, and other parts, such as the lower eyelid cannot be ignored. The other parts still interacts with the interface, which results in the differences of the four channels when in various oculogyria actions. Exactly, it is these disparate signals responding to the complex motion of periocular skin contributes to the decoding of eye motions.

As for the horizontal eyeball rotation, the periocular movement patterns are more complex, the upper and lower eyelid both interacts with the interface. we have added an approximate mechanism based on the signal responding to a right eyeball rotation. As is shown in an upward view (Supplementary Fig. 13b), the outlines of eyelid are depicted base on the Supplementary Fig. 11b and Movie 1. The periocular skin is

supposed positive charged to equilibrate the negative interface. From stage I to II, it is obviously that the center parts of upper and lower eyelid are away from the channel of VC1 and VC2, which cause the electrons flow out of the back electrodes resulting the negative voltage in VC1 and VC2. As for the area opposite to HC2, as shown in Supplementary Fig. 13c (a vertical view), when eyeball turning right, the right side of the lower eyelid is concave to close to HC2, but the right side of the upper is convex, and the final effect on HC2 is a negative potential signal. It is totally contrary of HC1, the convex lower eyelid and concave upper eyelid is observed and finally it results in a negative potential on HC1. Expressly, the actual interactions between periocular skin and the interface are beyond of this simplified illustration, Thus, the signal of dextroversion has some differences with that of laevoversion (Fig. 3c, Supplementary Fig. 14c and d). Besides, the number of electrons in the figure is just qualitatively describe the direction of the potential but not quantitatively. We have added these mechanisms in the Supplementary Note 1.

Supplementary Fig. 13 (b and c) Mechanism analysis of the interface array to right oculogyria with upward view (b) and vertical view (c).

7. In Supplementary Fig. 13, considering electron flows in VC2 at stage II~IV, the electric signals in HC1 and HC2 should be similar to those in VC2. However, different signals and intensities are shown in Fig. 3b

Answer) This is a very kind comment. Just as discussed above, there exist different interactions between the four channels, especially the channel of HC1 and HC2 which

are influenced simultaneously by the different movement patterns of the upper and lower eyelid. In addition, the number of electrons labeled in the figure is just qualitatively describe the direction of the potential but not quantitatively. So that, the signals in HC1 and HC2 is different with VC2, but nearly in the same trends. We further clarified this part, as can be seen on page 10.

8. The authors should clearly explain output signals in Fig. 3c and Supplementary Fig. 14. Based on the mechanism, the authors should explain the results in Supplementary Fig. 15~22.

Answer) Thanks for this suggestion. The high surface charge density of the bilayer contributes to the improved sensitivity of this AET system. Thus, we realize the eye-tracking sensitivity of 5° . Supplementary Fig. 14 shows the signal changed with various oculogyria angles, and there are slight differences between the signals of different angles of single direction of oculogyria because the small destabilization of eye movements. As for the signal the signal in Fig. 3c, it is collected without quantitative angle, but it is nearly the same as the signal of 30° in Supplementary Fig. 14. We further clarified this part, as can be seen on page 10.

About the signal in Supplementary Fig. 15~22, it can be seen as the half signal of that in Fig. 3d–e and Supplementary Fig. 14 c–d. Taking an example for verifying, as seen in the following figure, we compose the signal form Supplementary Fig. 15 and 16 into a signal, which is nearly the same as that in Fig. 3d (signal of 15°), because we have quantified the angle that is about 15° with a scale board (Supplementary Fig. 14) to collect the signal in Supplementary Fig. 15–22.

Revised Fig. 1 (b and c) Composed signal of the eye movement from 6→9 with 9→6.

9. The authors should introduce previous literatures on electric signal-based eye trackers. In the reference “Imperceptible electrooculography graphene

sensor system for human–robot interface” (*npj 2D Materials and Applications* 2018), the EOG sensor has been developed, which is responsive to the motion of eye blinking. In addition, the authors should introduce previous literatures on EOG sensors based on the triboelectric mechanism.

Answer) We apologized for missing references about EOG sensor. The EOG sensor based on the eye dipole with a positive cornea and negative retina, enables eye tracking even in total darkness, and is theoretically owns high resolution based on pairs of electrodes to detect the deflection of the corneo-retinal dipole and has been used in areas of human-machine interfaces and medical diagnosis³⁰. For example, Prof. Akinwande’s group has developed EOG sensors based on electronic tattoo applied in for eye-controlled quadcopter²⁷, and even realize resolution of 4° of eye tracking²⁸.

As another alternative choice, TENG has also been used for eye tracking in recent years⁴¹. Hu’s group proposed a TENG-based sensor for tracking blinks and used it for eye-controlled typing system⁴⁰. Different with reported electric signal-based sensor. We improved this AET system in a fully non-contact mode based on electrostatic interactions, which is a new progress in comparing with previous TENG-based eye sensor. We also developed several new functions for the TENG-based eye tracking, such as vision-based commercial analysis and eye-controlled input modality.

Thanks again for your worthy suggestion, and we have added the related discussions and references in the revised manuscript. We further clarified this part, as can be seen on page 2–3 and Supplementary Table 4, and the many new references about EOG and TENG-based trackers has been updated as following:

- [22]. Multipurpose and reusable ultrathin electronic tattoos based on PtSe₂ and PtTe₂. *ACS Nano* **15**, 2800-2811 (2021).
- [27]. Fabrication, characterization and applications of graphene electronic tattoos. *Nat. Protoc.* **16**, 2395-2417 (2021).
- [28]. Imperceptible electrooculography graphene sensor system for human–robot interface. *npj 2D Mater. Appl.* **2**, (2018).
- [30]. Highly stretchable starch hydrogel wearable patch for electrooculographic signal detection and human–machine interaction. *Small Struct.* **2**, (2021).
- [40]. Eye motion triggered self-powered mechnosensational communication system using triboelectric nanogenerator. *Sci. Adv.* **3**, e1700694 (2017).
- [41]. Triboelectric patch based on maxwell displacement current for human energy harvesting and eye movement monitoring. *ACS Nano* **16**, 11884-11891 (2022).

Reviewer #2 (Remarks to the Author):

This work uses the principle of non-contact electrostatic induction in a contactless triboelectric sensor, using an array of four sensors on the glasses. The authors use the eyelash for detection of real-time eye movements and using this concept they created eye-controlled human-computer interactions. The work is presented well, and results are satisfactory with details implementations for some unique applications. The design of sensor has been done in an optimum way to obtain maximum performance. Through Machine learning they are able to identify the movement of the eye using classification of the signals of the four sensors.

Comparing to the literature on eye tracking, some of them have been cited as well, I find the contributions is limited for this journal.

Answer) Thank you very much for approving the performance of this AET system and thank you for your valuable suggestion. Previously, major approaches in eye-tracking areas are based on electrical, optical and magnetic resonance (MR) signals. Compared with electrical tracker, MR-based tracker relies independently on cumbersome equipment, leading to insufficient, portability. Although optical method is notable for high resolution based on sclera-mirrored infrared light, the privacy concerns and the awkward location of the camera are disadvantages. Supplementary Table 3 has compared advantages and disadvantages of these main eye tracking methods. After that, the EOG and TENG-based tracker, as the most typical non-invasive electrical method, are summarized in Supplementary Table 4. This proposed AET system merit for its non-contact tracking mechanism, while EOG sensor depends highly on the skin-integrated pairs electrode that bring disadvantages in skin breath and infectious risk. Meanwhile, the glasses-integrated sensing interface endows this AET system with enhanced portability and wearability. Therefore, this AET system is an alternative eye-tracking choice applied for medical monitoring and human-computer interactions.

In our work, a charge-keeping material with the optimized dielectric bilayer and surface modification of Ag NWs electrode is proposed, where a breakthrough of surface charge density ($1671.10 \mu\text{C}\cdot\text{m}^{-2}$) and an ultrahigh charge-keeping rate of 96.91% can be obtained. The high surface charge density allows the charged interface to have an eye-tracking resolution of 5° , which is so far the highest value for non-contact eye-tracker based on electrical signal. In addition, this is the first time that a fully non-contact and transparent eye tracking system is realized by using electrostatic induction.

We have developed several interesting functions for eye tracking system, especially the function of closing-eye monitoring that cannot be easily realized with other optical trackers. Furthermore, the proposed preference analysis system and eye-controlled input modality containing signal acquisition with deep learning, has also promoted electrostatic sensing applied in commercial analyzing, medical monitoring and human-computer interactions.

Overall, this AET system is an interdisciplinary research that is quite suitable for the multidisciplinary scope of Nature Communications.

We further clarified this part (on page 13 and Supplementary Table 4).

It will be interesting to include the electronics used for data acquisition.

Answer) The signals origin from the electrostatic induction between periocular skin and the charged interface are triboelectric nanogenerator-based open-circuit voltages. As seen in the Methods part, we used a NI USB-6356 Multifunction I/O Device and LabVIEW software for data acquisition, and decoded the oculogyria-based characteristic signal with deep learning.

Thanks again for this kind review that is helpful to improve our manuscript.

Reviewer #3 (Remarks to the Author):

Eye tracking is a method of recording and analyzing eye movements to gain insights into a person's visual attention and thought processes. It is used in various fields, such as virtual reality, human-computer interaction, and medical monitoring, providing valuable information on visual attention, cognitive processes, and behavior. The authors of this manuscript propose a new type of active eye tracking (AET) system, based on the electrostatic induction effect. This system is designed to be transparent, flexible, and ultra-persistent, offering improved capabilities for monitoring and analyzing eye movements. The proposed AET system uses a triple-layer structure to enhance its performance. The structure consists of a dielectric bilayer and a rough-surfaced Ag nanowire electrode layer, which improves the intrinsic capacitance and charge storage capability of the electrostatic interface through increased interfacial trapping density. This advanced system offers oculogyric detection with an impressive angular resolution of 5° . The results in this study are robust and well-supported by comprehensive set of experimental data. The real-time decoding of eye movements, enabled by the AET systems, can be of great use for many envisioned applications. Below, please find several questions and comments which can help the authors improve the manuscript.

Q1. The authors utilized 4 channels in the interface array. Can the experimental or theoretical basis for determining this number of channels be discussed? Additionally, it would be helpful to have an explanation of how the performance of the system is affected as the number of channels increases or decreases.

Answer) Thanks for this kind suggestion. Following with the observation in Supplementary Movie 1, we adopt 3D face scan for analyzing the eye movement patterns. As we discussed in Fig. 3a, the vertical oculogyria induced the identical motion of the upper and lower eyelids. It is worth mentioning that the lower eyelid moves characteristically during horizontal oculogyria, that the skin fluctuates concavely and convexly with the contract and relaxation of the muscle respectively, and the upper eyelid is convex along with the eyeball's position. Based on these motion characteristics, we arrange VC1 and VC2 in front of the middle position of upper and lower eyelid, where are the most dynamic location during oculogyria. Regarding HC1 and HC2, their locations are at the two sides of VC2, where they can theoretically

induce maximum electrostatic interactions with the dynamic lower eyelid.

Additionally, we believe these four channels is set according with the investigated oculogyria patterns, and if channels are decreased, such as 3 channels in the following figure, it would increase difficulties to distinguish oculogyria directions. For example, signals of infraversion and dextroversion in Revised Fig. 2a, as well as supraversion and laevoversion in 2c is hard to differentiate. As for the increase of channels, we proposed a 6-channel model in Revised Fig. 2d, which can theoretically provide distinguished signals to decoding eye movement, but it will increase difficulties in data acquisition and computational analysis. Therefore, the simplified 4-channed interface array, with optimized location based on oculogyria patterns, is capable of decoding eye movement with great accuracy.

Revised Fig. 2. The interface array with decreased and increased channels. (a–c) Electrostatic signals of Signals with missed VC2 (a), HC1 (b) and HC2 (c). (d) the outline of an interface with increased channels.

Q2. In the field of object recognition, several models such as AlexNet and ResNet exist. Could you explain why VGGNet was selected over other models for this study?

Answer) Thanks for this kind question. AlexNet, ResNet, VGGNet are three kinds of CNN network, and they all own good feature expression ability. To our knowledge, they can both realize the classification of data decoding. The VGGNet used here is a 1DCNN model, which has advantage of smaller number of parameters, increased model non-

linearity and reduces training time as well as the responding time.

Q3. It is mentioned in line 345 that the AET system has the capability of real-time classification. Can the authors specify the latency and processing time of the system?

Answer) As we discussed above, the 1DCNN network used in this AET system smaller number of parameters, increased model non-linearity and reduces training time. With the optimized location array, the characteristic signals are obviously to decode eye movement patterns, which has reduced the complexity of the network with short training and processing time. To be specific, the processing time is also a factor that seriously depended on the hardware configuration of the computer. We have investigated our responding time (with a calculating code) of the eye-controlled system that is within 1.002 ms even with random negligible latency (0 ms), which indicates the accuracy of this system. We further clarified this part, as can be seen on page 12 and Supplementary Fig. 24d.

Supplementary Fig. 24. (d) Responding time from signal input to movement decoding (random action in Fig. 5c–d).

Q4. According to the information provided in Supplementary Table 3, the system can detect close eye tracking. Can the system also detect eye movements when the eyes are closed?

Answer) This is a very valuable question for the application of this AET system. The AET system works on the electrostatic interactions, and the fluctuations of periocular skin can induce electrostatic signals in the channels. Thus, the oculogyria of closing eye can also be detected. We have shown signals of the system responding to closing eye in

Supplementary Fig. 14e, which can also decode the direction of eyeball rotation indicating the ability of this AET system applied in monitoring rapid-eye-movement sleep and diagnosing oculogyria of vegetable patients. Another video material is also provided (Video 2), which can address this question.

Q5. The line 387 mentions that the initial electrostatic signals of the 4 channels were determined through 2 blinks. Can the authors provide more detail on this process? Specifically, what data was used to determine the optimal location for the channels?

Answer) We thanks for this kind question. Since the array is arranged with the optimal location to the eye, when adhering the interface, a center in in front of pupil (eyes looking forward) should be mask to calibrate the center of the interface. Then the glasses should be wore appropriately making the interface at the right position. Before activating the decoding system, 2 blinks action and the related signal is induced to approximately execute the amplitude of VH1 (about 0.45 V) from the signal window (the upper right window in Supplementary Movie 3 and 4). After that, the real-time tracking system is activated then enter running state. We further clarified this part detailly, as can be seen on page 15.

Q6. Can this system potentially detect the change of pupil size as well?

Answer) Thanks for this considerable question. The size of pupil is controlled by the smooth muscles at inner eyeball. However, the contract and relaxation of theses muscles cannot induce motions of periocular skin. So, that, it would not generate electrostatic interactions with the interface and cannot be detected bay this AET system. Fortunately, this question about pupil detecting points another direction in eye tracking area, and will encourage us to develop new method to tracking complex eye movements.

We thanks again for your kind comments that are great worthy for us to improve our manuscript.

REVIEWER COMMENTS

Reviewer #1 (Remarks to the Author):

Authors addressed all the comments raised by the reviewers. Now, the manuscript is proper for publication in Nature Communications.

Reviewer #3 (Remarks to the Author):

The authors have clarified and expanded upon their methods and results, providing additional data and analysis to support their findings. The revisions have resulted in a more comprehensive and coherent manuscript, with improved organization and clarity. The AET system has the potential to decode eye movements in real-time, offering a range of practical applications, making also positive impact on the fields.

Reviewer #4 (Remarks to the Author):

This paper presents a novel non-contact method for tracking eye movements using a triboelectric mechanism. The authors have investigated the sensitivity of the measurement by optimizing material characteristics such as nano-roughness and surface charge, which sets this study apart from others that use similar methodologies. The paper provides a clear description of the strategies employed and includes a comprehensive set of data. However, the authors' rationale for using flexible electrodes attached to rigid eyeglasses appears unclear. While the authors claim that flexibility allows for adaptation to different physical properties of subjects, such as distance between eyes, the significance of this feature is not well-explained. Therefore, a major revision is recommended, including a clearer explanation of the significance of the flexible electrodes and data demonstrating improved accuracy. If these issues are addressed, acceptance of the paper may be possible.

1. I believe the non-contact tracking method based on triboelectrics is vulnerable to environmental factors such as humidity, air flow from moving objects, and temperature. I suggest that the authors

provide additional data under these circumstances. This could be beneficial, particularly since the authors have emphasized the importance of resolution in the introduction and the obtained data (the angular resolution of 5 σ).

2. It is unclear if the resolution of 5 σ is sufficient for accurately detecting eye movements in real-world applications. Providing additional background scientific knowledge in the field of eye tracking and motion detection could be helpful.

3. I would recommend that the authors enhance the novelty of their work by providing future applications of using flexible, transparent, and ultra-persistent electrodes in this specific approach.

4. Minor comments:

- The term "rough bond" used by the authors in Line 112 is ambiguous. It could mean a tight bond to a rough surface or potentially refer to the adhesion between two surfaces with uneven or irregular textures, where the bond is not perfectly smooth or uniform. To avoid confusion, I would suggest replacing this term with clearer wording. In addition, it may be helpful to clarify the relationship between "tight interface" and "rough bond" in Line 225.

- Fig. 2F: I would recommend providing scientific details regarding why the authors consider 20 μm to be the optimal value.

- I would suggest that the authors avoid using arrows in Figure 2b, as this could give the impression that the values are interrelated.

- Typo:

line 154, better

line 241, Fig. 9e-g

line 235, optimize

line 289, Fig 3d and 3e

REVIEWER COMMENTS

Reviewer #1 (Remarks to the Author):

Authors addressed all the comments raised by the reviewers. Now, the manuscript is proper for publication in Nature Communications.

Answer) Thank you very much for your valuable suggestions.

Reviewer #3 (Remarks to the Author):

The authors have clarified and expanded upon their methods and results, providing additional data and analysis to support their findings. The revisions have resulted in a more comprehensive and coherent manuscript, with improved organization and clarity. The AET system has the potential to decode eye movements in real-time, offering a range of practical applications, making also positive impact on the fields.

Answer) Thank you very much for your valuable suggestions.

Reviewer #4 (Remarks to the Author):

This paper presents a novel non-contact method for tracking eye movements using a triboelectric mechanism. The authors have investigated the sensitivity of the measurement by optimizing material characteristics such as nano-roughness and surface charge, which sets this study apart from others that use similar methodologies. The paper provides a clear description of the strategies employed and includes a comprehensive set of data. However, the authors' rationale for using flexible electrodes attached to rigid eyeglasses appears unclear. While the authors claim that flexibility allows for adaptation to different physical properties of subjects, such as distance between eyes, the significance of this feature is not well-explained. Therefore, a major revision is recommended, including a clearer explanation of the significance of the flexible electrodes and data demonstrating improved accuracy. If these issues are addressed, acceptance of the paper may be possible.

1. I believe the non-contact tracking method based on triboelectrics is vulnerable to environmental factors such as humidity, air flow from moving

objects, and temperature. I suggest that the authors provide additional data under these circumstances. This could be beneficial, particularly since the authors have emphasized the importance of resolution in the introduction and the obtained data (the angular resolution of 5°).

Answer) Thank you very much for this valuable suggestion. In order to address this comment, we have done a series of additional experiments to investigate the stability of the interface under inclement environmental factors. The experimental devices are shown in Supplementary Fig. 11a, and the open-circuit voltage of the interface under non-contact mode is collected under various humidity (I), air flow (II) and temperature (III). The detailed experiments are described in Methods part on Page 16.

Supplementary Fig. 11. Stability test of the interface under environmental factors. (a) Experimental devices for testing the stability of the interface under various humidity (I), air flow (II) and temperature (III). (b and c) Open-circuit voltage of the interface under increased humidity (b) and air flow (c). (d) Open-circuit voltage of the interface under heating and cooling conditions.

The influences of humidity are shown in Fig. 11b, where the voltage shows slightly

decrease until the humidity increased to nearly 90%. As for the air flow, it imposes non-interferences on the output performance of the interface. On the other hand, we have observed some interesting phenomenon during the temperature experiments, which is worthy to be analyzed. The voltage signal keeps stable under 40 °C, but increases (from 22.34 to about 51.95 V) within the heating progress from 44.2 to 100.2 °C. Moreover, the voltage signal return to 27.00 V, when the temperature decreases to 27 °C. It has been found that the negative-charged interface can attract positive ions (or particles) in the air (*Energy Environ. Sci.*, 2016, 9, 3085-3091). Under the hot air flow, these ions may be blow away or excited to the air, which causes the increased output voltage (the negative charge injected in the film is much more stable). Then, the re-attraction of positive ions can happen during cooling progress, leading to the decrease of voltage signal. Theoretically, the increased output performance is beneficial to improve the resolution of the electrostatic interface, which means our device can work more effectively with higher temperature. Meanwhile, considering surrounding environmental temperature for human rarely exceeds 40 °C, the device can maintain stable operation without the influence from temperature. Thanks for the author's comment, which help us to father elucidate the output performance of this electrostatic sensors, and we will further study the inherent relation between hot wind and contactless electrostatic sensors in the future. Finally, the available range (including factors of humidity, air flow and temperature) of this interface is summarized in Supplementary Table 3 with blue background. We further clarified this part, as can be seen on Page 8.

Furthermore, the eye-tracking performance of the interface are also studied under sweating condition. As shown in Supplementary Fig. 16, the periocular humidity increases from 42.7% to about 71.0% after sweating, which is a humidity range that shows no influence to the stability of the interface. In this case, the blink (Supplementary Fig. 16b and c) and supraversion (Supplementary Fig. 16d and e) signals are collected, which both show quite stable after sweating, indicating the stability of this eye-tracking system (on Page 11).

Supplementary Fig. 16. Eye-tracking performance under sweating. (a) Periocular humidity test after sweating. (b and c) Blink signals under nonsweating (b) and (c) sweating conditions. (d and e) Supraversion signals under nonsweating (d) and (e) sweating conditions.

Humidity (%)	Air flow (m/s)	Temperature (°C)
100	10	100
90	9	90
80	8	80
70	7	70
60	6	60
50	5	50
40	4	40
30	3	30
20	2	20
10	1	10
0	0	0

Stable operation range

Supplementary Table 3. The available range (including factors of humidity, air flow and temperature) of the electrostatic interface.

2. It is unclear if the resolution of 5° is sufficient for accurately detecting eye movements in real-world applications. Providing additional background scientific knowledge in the field of eye tracking and motion detection could be helpful.

Answer) Thanks for this important comment. It is undeniable that the smaller angle resolution is realized, the better performance of eye tracker is considered, and precise high resolution is always the important aims for researchers in eye-tracking areas. However, the smaller eye movement is difficult to be reliably measured by eye trackers (*Behav. Res.*, 2020, 52, 2098–2121). Thus, researchers always ensure that that stimuli on screen subtend exactly 5 degrees of visual angle (Visual Angle - Fast, Accurate, Reliable Eye Tracking), because the visual angle is calculated through triangle trigonometry with the shifting of gaze point and distance between eye and screen (Fig. R1). Meanwhile, as we summarized in Supplementary Table 4, the angle resolution of 5° is the smallest that realized by electrical-signal-based trackers, and we believe with ambitions targeting high resolution, the smaller oculogyria resolution will be realized.

Fig. R1. Calculation of visual angle.

3. I would recommend that the authors enhance the novelty of their work by providing future applications of using flexible, transparent, and ultra-persistent electrodes in this specific approach.

Answer) We thanks for this kind recommendation. We fabricated the flexible, transparent and ultra-persistent electrostatic interface for realizing active eye tracking. The adhesive PDMS substrate provides self-adhesion to glasses and the flexibility (both electrode and dielectric bilayer) allows the interface to be adhered on glasses with various curvature (Supplementary Fig. 13c), which is convenient to be adopted to diverse glasses such as protective glasses, sun glasses, myopia glasses of different

degrees as well as some flexible glasses (e.g. AirSpecs Tony Morgan London TM 1003/002S, Vision Express). In addition, the interface can also be stretched to the appropriate size (the distance between four electrodes) that adapts to various individuals (Supplementary Fig. 13d, the electrode distance is extended with 4 mm), and the ultra-persistent surface charges can guarantee the stable operation of the eye tracking system. Meanwhile, the well transparency of the interface is benefit for free eyesight which is advantageous for normally wearing these mentioned glasses. We further clarified this part, as can be seen on Page 10.

As for the application of this electrostatic interface, we have demonstrated its usage in visual favorite analysis for commercial marketing, and an eye-controlled input modality for the amyotrophic lateral sclerosis (ALS) patients when conducting human-computer interaction. Since the interface is also sensitive to closed-eye movement, it is supposed to be applied in monitoring rapid eye movement (REM) sleep. Furthermore, the eye tracking for patients with vegetative state (VS) is a potential application since eye movement is important clinical features (*Postgrad. Med. J.*, 1999, 75, 321). This active electrostatic interface equipped with soft glasses is a potential approach for clinical diagnosis of VS patients (Supplementary Fig. 27a). Besides, as the interface is based on the electrostatic induction with skin fluctuation, it can also be used for other skin-based sensing, such as monitoring the movement of larynx (Supplementary Fig. 27b), which is related with swallowing diseases. Except body sensing, this electrostatic can also realize contactless human-machine interactions to avoid direct contact in some cleaning environment (Supplementary Fig. 27c). We further clarified this part on Page 13.

Supplementary Fig. 13. (c) The flexible interface adhered on glasses with various curvature. (d) The stretched interface adhered on a pair of glasses.

Supplementary Fig. 27. Potential application of the electrostatic interface. (a) REM sleep monitoring. (b) Larynx movement monitoring. (c) Contactless human-machine interactions.

4. Minor comments:

- The term "rough bond" used by the authors in Line 112 is ambiguous. It could mean a tight bond to a rough surface or potentially refer to the adhesion between two surfaces with uneven or irregular textures, where the bond is not perfectly smooth or uniform. To avoid confusion, I would suggest replacing this term with clearer wording. In addition, it may be helpful to clarify the relationship between "tight interface" and "rough bond" in Line 225.
- Fig. 2F: I would recommend providing scientific details regarding why the authors consider 20 μm to be the optimal value.
- I would suggest that the authors avoid using arrows in Figure 2b, as this could give the impression that the values are interrelated.
- Typo: line 154, better; line 241, Fig. 9e-g; line 235, optimize; line 289, Fig 3d and 3e

Answer) Thanks for these valuable comments for improving our manuscript.

(1) The rough bond in Line 112 is indeed to describe the tight adhesion between the layers, we have replaced with "rough surface" (on Page 4). We have revised line 225 as "it will be cured on the rough Ag NWs and generate a tight layered structure" on Page 8.

(2) Fig. 2f shows the transferred charges of the dielectric bilayer with various thickness of PDMS on plate Cu electrode (the left 5 lines) which shows the highest value when PDMS set at 20 μm which is considered as the optimal thickness. Then the bilayer is placed on rough Ag NWs and obtained a higher transferred charge (the right line, labeled 20 rough Ag NWs in Fig. 2f), which indicates the advantages of the dielectric bilayer with rough surface.

(3) The arrows in Fig. 2b is removed in the revised manuscript.

Fig. 2b. Average surface charge density of PTFE, PVC and PCTFE.

(4) The mentioned typo has been revised in the related lines and the manuscript has been checked thoroughly.

REVIEWERS' COMMENTS

Reviewer #4 (Remarks to the Author):

The authors have presented a sequence of supplementary investigations that have yielded promising results regarding the negligible influence of environmental factors, such as sweating, as well as the novelty of implementing flexible electrodes. They have also demonstrated significant applicability, specifically the remarkable adhesion capabilities on diverse glass surfaces. As a reviewer, I am satisfied that the authors have diligently attended to all the issues I raised, and as a consequence, I am pleased to endorse the suitability of this manuscript for publication in Nature Communications.

REVIEWERS' COMMENTS

Reviewer #4 (Remarks to the Author):

The authors have presented a sequence of supplementary investigations that have yielded promising results regarding the negligible influence of environmental factors, such as sweating, as well as the novelty of implementing flexible electrodes. They have also demonstrated significant applicability, specifically the remarkable adhesion capabilities on diverse glass surfaces. As a reviewer, I am satisfied that the authors have diligently attended to all the issues I raised, and as a consequence, I am pleased to endorse the suitability of this manuscript for publication in Nature Communications.

Answer) Thank you very much for your comments.